# Residential energy use emissions dominate health impacts from exposure to ambient particulate matter in India

Luke Conibear [1,2], Edward W. Butt[2], Christoph Knote [3], Stephen R. Arnold[2] & Dominick V. Spracklen[2]

Exposure to ambient fine particulate matter ($PM_{2.5}$) is a leading contributor to diseases in India. Previous studies analysing emission source attributions were restricted by coarse model resolution and limited $PM_{2.5}$ observations. We use a regional model informed by new observations to make the first high-resolution study of the sector-specific disease burden from ambient $PM_{2.5}$ exposure in India. Observed annual mean $PM_{2.5}$ concentrations exceed $100\,\mu g\,m^{-3}$ and are well simulated by the model. We calculate that the emissions from residential energy use dominate (52%) population-weighted annual mean $PM_{2.5}$ concentrations, and are attributed to 511,000 (95UI: 340,000–697,000) premature mortalities annually. However, removing residential energy use emissions would avert only 256,000 (95UI: 162,000–340,000), due to the non-linear exposure–response relationship causing health effects to saturate at high $PM_{2.5}$ concentrations. Consequently, large reductions in emissions will be required to reduce the health burden from ambient $PM_{2.5}$ exposure in India.

[1] Engineering and Physical Sciences Research Council (EPSRC) Centre for Doctoral Training (CDT) in Bioenergy, University of Leeds, Leeds, LS2 9JT, UK. [2] Institute for Climate and Atmospheric Science, School of Earth and Environment, University of Leeds, Leeds, LS2 9JT, UK. [3] Meteorological Institute, Ludwig-Maximilians-University Munich, Theresienstr. 37, 80333 Munich, Germany. Correspondence and requests for materials should be addressed to L.C. (email: pmlac@leeds.ac.uk)

Exposure to ambient fine particulate matter (PM$_{2.5}$) is a leading risk factor to human health. India experiences high annual mean ambient PM$_{2.5}$ concentrations, of up to 150 μg m$^{-3}$ across the Indo-Gangetic Plain (IGP)[1], where more than 50% of the country's population lives[2]. The Global Burden of Diseases, Injuries, and Risk Factors Study 2016 (GBD2016) estimated that one-quarter of the global deaths attributed to ambient PM$_{2.5}$ exposure occur in India[3–5]. Estimates of premature mortality from exposure to ambient PM$_{2.5}$ in India vary by a factor of 3, between 392,000 to 1,090,000 per year[3,5–17], with differences due to variations in ambient PM$_{2.5}$ estimates, health functions, population data sets and methodological approaches. Previous studies using global models, at a relatively coarse resolution, may not resolve PM$_{2.5}$ concentrations over polluted areas resulting in an underestimate of air pollution related premature mortality[18]. Evaluation of simulated PM$_{2.5}$ across India has also been limited, with extensive surface observations of Indian PM$_{2.5}$ only becoming publicly available in 2016. Understanding the contribution of different emission sectors to ambient air pollution is needed for effective pollution abatement efforts. In contrast to Europe and the USA, where air pollutant emissions from energy, industry, agriculture and land transport dominate, over India emissions from residential energy use are substantial[13,15,17,19]. Over half of India's population use solid fuels for their energy needs, and this emission sector makes an important contribution to ambient PM$_{2.5}$[13,20–22]. Previous global studies have estimated that emissions from residential energy use cause 73,000 to 460,500 premature mortalities across India each year[13,15,17,20,22].

There are two main methods of estimating the sectoral contributions to premature mortality from ambient PM$_{2.5}$ exposure, each giving greatly different results[23]. The subtraction (or zero-out) method calculates the sector-specific mortality as the difference between the all-source premature mortality estimate and a premature mortality estimate based on a model simulation where the emission sector has been removed[15,23,24]. Alternatively, the attribution method calculates sector-specific mortality as the sectoral fractional contribution to PM$_{2.5}$ concentrations multiplied by the total premature mortality estimate[13,17,20,23,25,26]. The non-linear exposure–response relationship means these two methods give different estimates, particularly for regions with high PM$_{2.5}$ concentrations such as India[27]. The two methods also

answer different questions: the attribution method estimates the number of premature mortalities that could be attributed to a sector's emissions, while the subtraction method estimates the reduction in premature mortalities that could be achieved by removing sector emissions.

Here, we use a regional numerical weather prediction model online-coupled with atmospheric chemistry at 30 km horizontal resolution to study the disease burden due to ambient PM$_{2.5}$ exposure in India from seven emission sectors. We performed a control simulation with all emission sources for the year of 2014, then annual sensitivity simulations were performed removing the respective emissions from each source sector of agriculture (AGR), biomass burning (BBU), dust (DUS), power generation (ENE), industrial non-power (IND), residential energy use (RES) and land transport (TRA). This study is the first to use high-resolution online-coupled simulations to estimate the contribution of different emission sectors to ambient PM$_{2.5}$ concentrations and related disease burden from exposure across India. We find that removing emissions from residential energy use causes the largest reduction in ambient PM$_{2.5}$ exposure in India and has the greatest benefit to human health.

## Results

**Evaluation of surface PM$_{2.5}$.** Figure 1 compares simulated and observed surface PM$_{2.5}$ concentrations over India. The model simulates high annual mean PM$_{2.5}$ concentrations (>100 μg m$^{-3}$) over the IGP, and lower concentrations over central and southern India, broadly matching observations (Fig. 1a). Overall, the model is unbiased against observed annual mean PM$_{2.5}$ abundances (normalised mean bias, NMB = −0.10) (Fig. 1b). Comparison of simulated aerosol optical depth (AOD) against the aerosol robotic network (AERONET) (Supplementary Fig. 1) shows similar agreement (NMB = 0.09). The model has limited success in reproducing the spatial variability of PM$_{2.5}$, underestimating near the Thar desert and in the central IGP. Underestimation of dust in the western IGP has been identified previously[28] and likely contributes to model underestimation of PM$_{2.5}$ in this region. Simulated PM$_{2.5}$ concentrations are greatest in winter (DJF) and autumn (SON) and lowest in spring (MAM) and summer (JJA), matching observations (Fig. 1b, Supplementary Fig. 2). Simulated population-weighted annual mean PM$_{2.5}$ concentration across

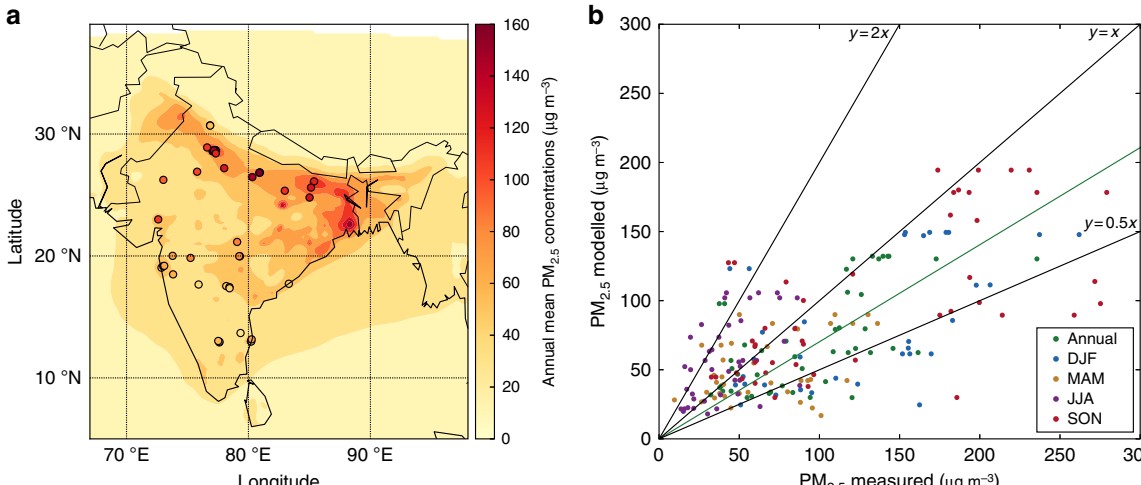

**Fig. 1** Comparison of observed and simulated PM$_{2.5}$ concentrations. **a** Annual mean surface PM$_{2.5}$ concentrations. Model results for 2014 (background) are compared with surface measurements from 2016 (filled circles). **b** Comparison of annual and seasonal mean surface PM$_{2.5}$ concentrations. The best fit line (green), 1:1, 2:1 and 1:2 lines are shown (black). Annual, winter (DJF), spring (MAM), summer (JJA) and autumn (SON) normalised mean bias (NMB) are −0.10, −0.24, −0.07, 0.69 and −0.10, respectively. The best fit line for annual data has slope = 0.70 and Pearson's correlation coefficient ($r$) = 0.19

**Table 1 Reduction in population-weighted annual mean PM$_{2.5}$ concentrations in India caused by removing different emission sectors**

|  |  | AGR | BBU | DUS | ENE | IND | RES | TRA |
|---|---|---|---|---|---|---|---|---|
| Reduction to population-weighted PM$_{2.5}$ (PM$_{2.5\_SECTOR\_OFF}$) | µg m$^{-3}$ | 0.2 | 1.6 | 0.0 | 12.0 | 9.3 | 29.5 | 5.9 |
|  | % | 0 | 3 | 0 | 21 | 16 | 52 | 10 |

Sectors are agriculture (AGR), biomass burning (BBU), dust (DUS), power generation (ENE), industrial non-power (IND), residential energy use (RES) and land transport (TRA)

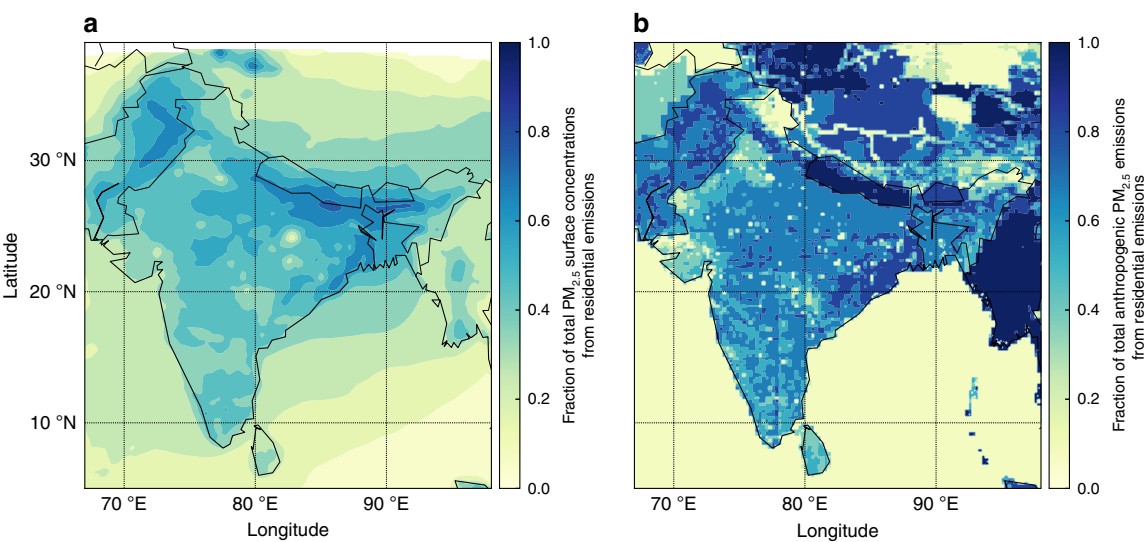

**Fig. 2** Fractional contribution of residential energy use to annual mean PM$_{2.5}$. **a** Concentrations. **b** Anthropogenic emissions. Emissions are from EDGAR-HTAP v2.2 (see Methods)

India was 57.2 µg m$^{-3}$. Our model simulations show that in 2014, 99% of the Indian population was exposed to annual mean PM$_{2.5}$ concentrations that exceeded the World Health Organization (WHO) Air Quality Guideline (AQG) of 10 µg m$^{-3}$ and 81% above the WHO Level 1 Interim Target (IT-1) of 35 µg m$^{-3}$.

**Contribution of emission sectors to ambient PM$_{2.5}$.** To investigate the contribution of different emission sectors to ambient PM$_{2.5}$ over India, we switched off emissions from different sectors one at a time in individual annual simulations. Table 1 shows the contribution of the different emission sectors to annual mean PM$_{2.5}$ concentrations across India.

The largest reductions in population-weighted ambient PM$_{2.5}$ concentrations are achieved through the removal of emissions from residential energy use (52%) followed by power generation (21%), industry (16%) and land transport (10%). Removing emissions from residential energy use reduces ambient PM$_{2.5}$ concentrations by as much as 70% over the IGP, with reductions of 30–50% over southern India (Fig. 2a). Residential energy use contributes 67% of annual anthropogenic PM$_{2.5}$ emissions across India, with contributions as great as 90% over the IGP (Fig. 2b). Residential energy use across India is an important PM$_{2.5}$ source throughout the year contributing 62% of anthropogenic emissions of PM$_{2.5}$ in summer and 70% in winter (Supplementary Fig. 3). Emissions from residential energy use contribute a larger fraction of anthropogenic PM$_{2.5}$ emissions than to ambient PM$_{2.5}$ concentrations due to non-anthropogenic sources of PM$_{2.5}$, including dust and biomass burning, and due to the contribution of aerosol precursors such as SO$_2$ and NO$_x$, for which industry

and power generation dominate emissions. Source apportionment suggests that 46–73% of BC concentrations in India are from non-fossil source (residential biofuel and biomass burning)[22], which broadly matches our estimate of the contribution of residential emissions to PM$_{2.5}$ concentrations.

**Premature mortality due to ambient PM$_{2.5}$ exposure.** We estimate total premature mortality due to exposure to ambient PM$_{2.5}$ in India as 990,000 (95% uncertainty interval (95UI): 660,000–1,350,000) per year, with 24,606,000 (95UI: 14,567,000–32,698,000) years of life lost (YLL). Most premature mortality due to exposure to ambient PM$_{2.5}$ occurs in urban areas (76%), defined by regions with population density larger than 400 persons km$^{-2}$. The spatial distribution of disease burden is shown in Fig. 3a. The IGP accounts for 71% of the premature mortalities associated with ambient PM$_{2.5}$ exposure with the dominant state (Uttar Pradesh) contributing 19%. The disease burden attributable to exposure to ambient PM$_{2.5}$ is dominated by ischaemic heart disease (IHD) (35%) and chronic obstructive pulmonary disease (COPD) (31%).

Figure 4 compares estimated premature mortality due to exposure to ambient PM$_{2.5}$ in India from this study with previous studies. The estimated premature mortality in India due to ambient PM$_{2.5}$ exposure estimated in this study agrees to within 4% with that from the GBD2015[3,4] and GBD2016[5] which had similar PM$_{2.5}$ concentrations (Supplementary Fig. 4), but is up to a factor of 2 greater than in many other studies. These previous studies applied different PM$_{2.5}$ concentrations at a range of spatial resolutions (0.1°–2.8°), as well as different population data sets,

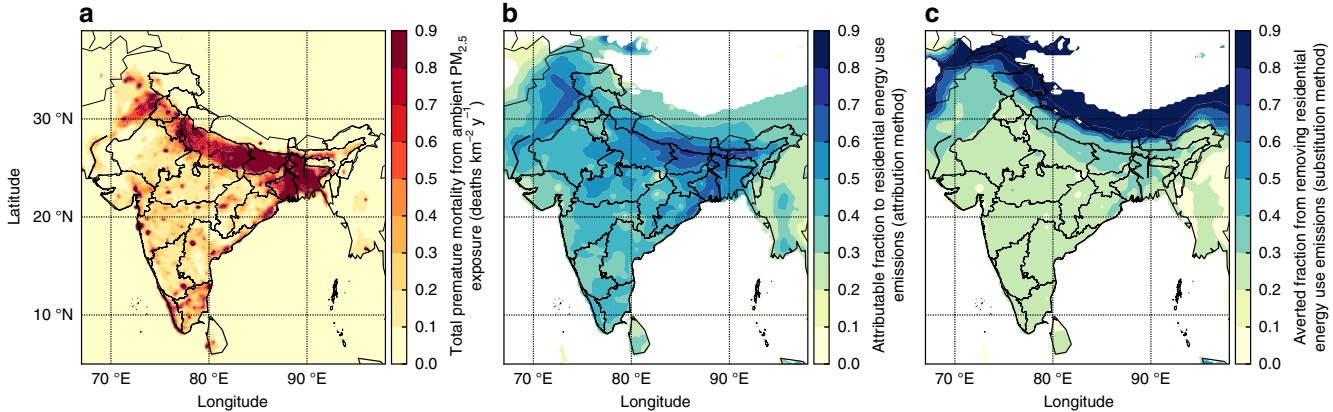

**Fig. 3** Disease burden due to exposure to ambient PM$_{2.5}$ across India. **a** Estimate of annual premature mortality due to exposure to PM$_{2.5}$ in India. **b** Attributable fraction of premature mortalities from residential energy use emissions (attribution method). **c** Averted fraction of premature mortalities from removing residential energy use emissions (substitution method)

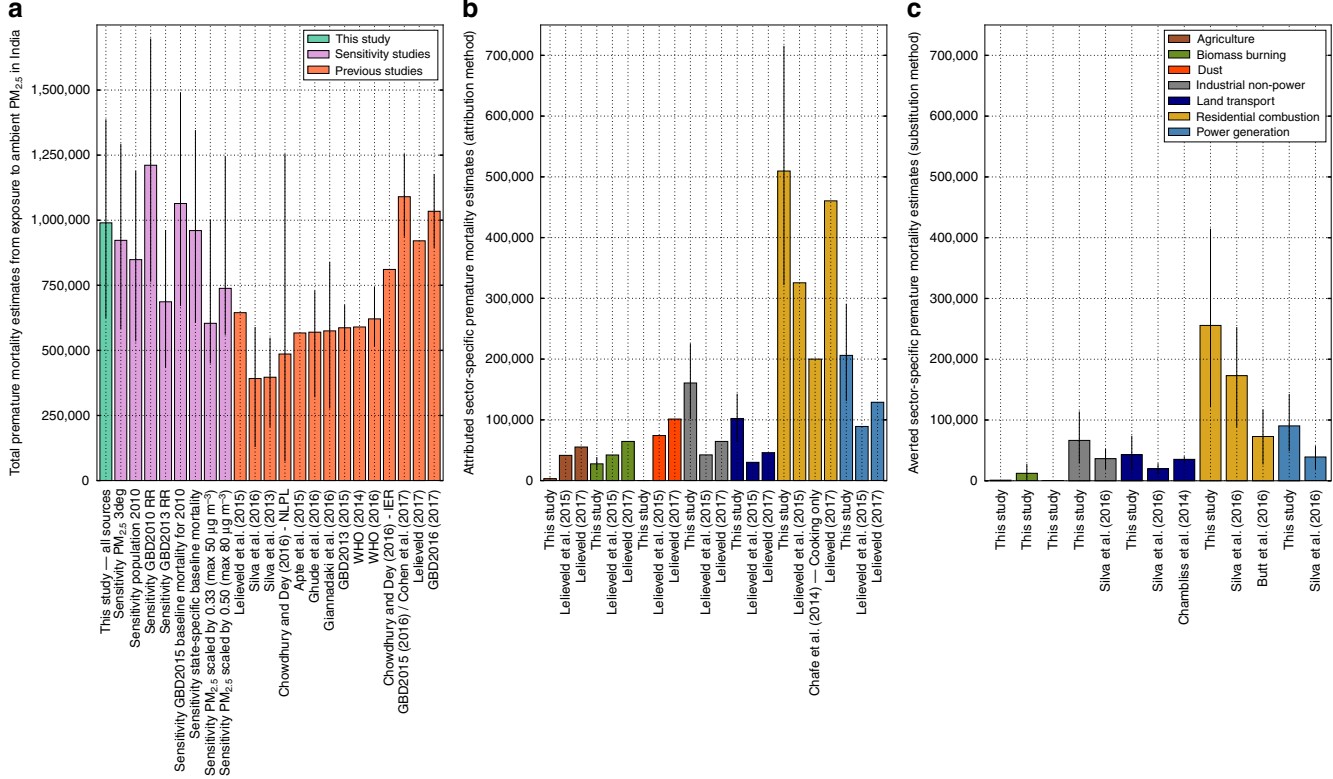

**Fig. 4** Comparison of annual premature mortality estimates for India due to exposure to ambient PM$_{2.5}$. **a** Total annual premature mortality from exposure to ambient PM$_{2.5}$ from all emission sources. This study (green) and sensitivity studies (purple) comparing varying model spatial resolution, population year, exposure–response function, baseline mortality rates and PM$_{2.5}$ concentrations are compared with previous studies (orange). **b** Attributed sector-specific estimates from the attribution method from this study compared to previous studies. **c** Averted sector-specific estimates from the substitution method from this study compared to previous studies. Error bars for this study and sensitivity analyses represent 95% uncertainty intervals (95UI) calculated from combining fractional errors in quadrature (see Methods). Error bars for previous studies given at the 95% uncertainty level where provided

exposure–response functions and baseline mortalities, all of which may play a role in the different premature mortality estimates. We explored likely reasons for lower premature mortality estimates in many previous studies and summarised the results in Fig. 4. Using lower resolution PM$_{2.5}$ data (3° in place of 0.3°) reduced population-weighted PM$_{2.5}$ concentrations by 20% but reduced our premature mortality estimate by only 7%, due to the non-linear exposure–response relationship

(Supplementary Fig. 5). We note this approach does not account for the effect of resolution on the representation of atmospheric processes within the model[18]. Using population data for 2010 (SEDAC GPWv4 UN-adjusted), when the Indian population was 7% lower than in 2015, reduced our premature mortality estimates by 14%. Using the relative risk (RR) from the GBD2010 integrated-exposure response (IER) function[9,29], as in a number of previous studies[7,9–11,13,15], increased our premature

| | | | AGR | BBU | DUS | ENE | IND | RES | TRA |
|---|---|---|---|---|---|---|---|---|---|
| **Table 2 Estimated premature mortality associated with ambient PM$_{2.5}$ exposure in India** | | | | | | | | | |
| Subtraction method | Premature mortalities per year ($M_{SECTOR}$) | Number (×10$^3$) | 1 (1–1) | 12 (8–16) | 0 (0–0) | 90 (60–122) | 66 (45–90) | 256 (162–340) | 43 (29–58) |
| | | % | 0 | 1 | 0 | 9 | 7 | 26 | 4 |
| Attribution method | Premature mortalities per year ($M_{SECTOR}$) | Number (×10$^3$) | 3 (2–5) | 28 (18–38) | 0 (0–0) | 208 (139–283) | 161 (107–220) | 511 (340–697) | 102 (68–139) |
| | | % | 0 | 3 | 0 | 21 | 16 | 52 | 10 |

Results are shown for different emission sectors (both absolute and fractional) for both subtraction and attribution methods. Values in parentheses represent the 95% uncertainty intervals (95UI). Sectors are agriculture (AGR), biomass burning (BBU), dust (DUS), power generation (ENE), industrial non-power (IND), residential energy use (RES) and land transport (TRA)

mortality estimates by 22%. In contrast, applying RR from GBD2013[12] reduced our mortality estimate by 31%. Using the GBD2015 baseline mortality estimates for 2010[63] increased our premature mortality estimates by 7%. Applying state-specific baseline mortality values[7] (see Methods) reduced our estimate of premature mortality by 3%, while increasing impacts over the IGP due to the higher baseline mortality rates in this region (Supplementary Fig. 6). Many previous studies[9,11,12,15,20,30–32] estimated maximum annual mean PM$_{2.5}$ concentrations across the IGP to be between 50 and 80 µg m$^{-3}$, but lacked widespread measurements of PM$_{2.5}$ for model evaluation. New observations suggest annual mean PM$_{2.5}$ concentrations of at least 160 µg m$^{-3}$ across the IGP, which is well simulated by our model. Scaling our PM$_{2.5}$ concentrations so that annual mean concentrations do not exceed 80 and 50 µg m$^{-3}$ reduced our premature mortality estimates by 25% and 39%, respectively. Overall, our analysis suggests that different exposure–response relationships and different PM$_{2.5}$ concentrations cause the largest differences in premature mortality estimates for India and are likely driving the majority of the differences between previous estimates.

**Contribution of emission sectors to disease burden**. Table 2 shows the premature mortality estimates per emission sector for both the subtraction and attribution methods (see Methods, Supplementary Table 1 shows disease-specific results and Supplementary Table 2 shows sector-specific YLL). The premature mortality estimates from the subtraction method (Fig. 3c) are a factor of 2–2.5 smaller than the attribution method (Fig. 3b). This is due to the non-linear exposure–response relationship, where health effects saturate at high PM$_{2.5}$ concentrations[23,27]. Our estimate of the reduction in premature mortality through removing an emission sector (subtraction method) is therefore a factor 2–2.5 times less than the estimate of the premature mortality attributed to that sector (attribution method). Consequently, the summation of sector contributions from the subtraction method is 469,000 (95UI: 304,000–626,000) premature mortalities per year (47% of control), which is substantially lower than the sector summation from the attribution method of 1,012,000 (95UI: 675,000–1,381,000) premature mortalities per year (102% of control). Overall, this has implications for attempts to reduce air pollution mortality in regions with high PM$_{2.5}$ concentrations.

For both methods, residential energy use is the dominant contributor to premature mortalities due to exposure to ambient PM$_{2.5}$ across all states in India, except for Delhi where emissions from land transport are dominant. We estimate that emissions from residential energy use cause 511,000 (95UI: 340,000–697,000) premature mortalities per year, 52% of the total premature mortalities due to ambient PM$_{2.5}$ exposure (attribution method). Removing emissions from residential energy use would prevent 256,000 (95UI: 162,000–340,000)

premature mortalities per year, 26% of the total premature mortalities due to ambient PM$_{2.5}$ exposure (subtraction method). After residential energy use, the next largest contributions are from power generation (9% of total premature mortalities for subtraction method and 21% for attribution method), industry (7% for subtraction and 16% for attribution) and land transport (4% for subtraction and 10% for attribution).

Figure 4 compares the estimates of the source-specific premature mortality from ambient PM$_{2.5}$ exposure in India. Previous studies also find emissions from residential energy use to dominate the contribution to PM$_{2.5}$ exposure-associated premature mortality in India[13,15,17]. Power generation was the next largest contributor in all studies[13,15,17], while the percentage contribution in this study is approximately a factor of 2 larger. Industrial emissions was third largest in both this study and the previous study using the subtraction method[15], while dust was third for the studies using the attribution method[13,17]. The percentage contribution from land transport was double that of all previous studies[13,15,17]. The two previous studies[13,15] that used the GBD2010 RR (which increased our estimates by 22%) obtained substantially lower total premature mortality than our estimates, likely due to lower PM$_{2.5}$ concentrations. More recent previous studies[3–5,17], using the GBD2015 RR, estimate similar total premature mortality to this study. Other studies estimating the contribution from residential energy use emissions to premature mortality are lower than this study due to a combination of using a log-linear exposure response function with lower PM$_{2.5}$ concentrations[22] or only estimating the contribution from residential cooking[20]. Our estimate of the annual number of premature mortalities attributed to residential energy use (511,000; 95UI: 340,000–697,000) is at the upper end of the range (73,000–460,500) from previous work[13,15,17,20,22].

In this study, we use a regional numerical weather prediction model online-coupled with chemistry to make the first high-resolution study of the contributions of seven emission sectors to the disease burden associated with ambient PM$_{2.5}$ exposure in India. New observations suggest that the annual mean PM$_{2.5}$ concentrations exceed 100 µg m$^{-3}$ across northern India, matching concentrations simulated by the model and confirming the conclusions of recent studies with similar PM$_{2.5}$ concentrations[4,10]. Sensitivity studies suggest that different exposure–response relationships and PM$_{2.5}$ concentrations drive the largest differences in estimates of premature mortality for previous studies. We find that residential energy use contributed 52% of population-weighted annual mean PM$_{2.5}$ concentrations resulting in an estimated 511,000 (95UI: 340,000–697,000) premature mortalities per year. We estimate that completely removing residential emissions would prevent 256,000 (95UI: 162,000–340,000) premature mortalities each year, 26% of the total premature mortalities due to exposure to ambient PM$_{2.5}$. The smaller relative reduction in premature mortality compared to the reduction in PM$_{2.5}$ concentration is due to the non-linear

exposure–response relationship, where the mortality response to $PM_{2.5}$ concentrations is sub-linear at the high $PM_{2.5}$ concentrations over India. Consequently, large reductions in emissions and $PM_{2.5}$ concentrations will be required to reduce the substantial health burden. Information on the source contributions to the burden of disease attributable to ambient $PM_{2.5}$ exposure is critical to support the national and sub-national control of air pollution.

## Methods

**Model description**. This study used the Weather Research and Forecasting model coupled with Chemistry (WRF-Chem) version 3.7.1[33]. WRF-Chem is fully online-coupled with modules for gas-phase chemistry and aerosol physiochemical processes, meaning the air quality component of the model is fully consistent with the meteorological component in using the same transport, grid coordinates, sub-grid scale physics and timestep. The Advanced Research WRF (ARW) solver used is fully compressible, non-hydrostatic and has an Eulerian mass conserving dynamical core[34]. Gas-phase chemical reactions are calculated using the chemical mechanism Model for Ozone and Related Chemical Tracers, version 4 (MOZART-4)[35], with several updates to photochemistry of aromatics, biogenic hydrocarbons and other species relevant to regional air quality[36,37]. Photolysis rates are calculated with the Fast Tropospheric Ultraviolet–Visible (FTUV) module[38]. Aerosol physics and chemistry are represented by the Model for Simulating Aerosol Interactions and Chemistry (MOSAIC) scheme, with no sub-grid convective aqueous chemistry[39,40], and the Kinetic PreProcessor (KPP)[41]. Four sectional discrete size bins are used within MOSAIC: 0.039–0.156 μm, 0.156–0.625 μm, 0.625–2.5 μm, 2.5–10 μm. The Thompson scheme was used for cloud microphysics[42] and the Grell 3-D scheme for convective parameterisation[43]. The Rapid Radiative Transfer Model (RRTM) option is used for both short and long wave radiation[44]. Simulated mesoscale meteorology is kept in line with analysed meteorology through grid nudging to the National Centre for Environmental Prediction (NCEP) Global Forecast System (GFS) analyses to limit errors in mesoscale transport[45,46]. The model meteorology was reinitialised every month to avoid drifting of WRF-Chem, while chemistry and aerosol fields were kept to allow for pollution buildup and mesoscale transport phenomena to be captured. During the simulations, horizontal and vertical wind, potential temperature and water vapour mixing ratio were nudged to GFS analyses in all model layers above the planetary boundary layer. Meteorological conditions were initialised by NCEP GFS 6-hourly analyses at 0.5° resolution. These, together with GFS 3-h forecasts in between were also used for boundary conditions and grid analysis nudging[45,46]. MOZART-4/Goddard Earth Observing System Model version 5 (GEOS5) 6-hourly simulation data were used for chemical and aerosol boundary conditions[47]. The regional model domain (Supplementary Fig. 7) is represented as a 140×140×34 cell grid on a Lambert conformal conical projection with a horizontal resolution of 30 km, extending vertically up to 10 hPa. The simulation period was for the year of 2014. Model setup is detailed in Supplementary Table 3.

Simulated $PM_{2.5}$ mass concentrations over India using WRF-Chem have recently been evaluated against ground observations[10]. The gas-phase chemistry scheme used in this study (MOZART-4) has been used in other studies over India and captured important observed features of gas-phase species[10,28]. These studies[10,28] used the Georgia Tech/Goddard Global Ozone Chemistry Aerosol Radiation and Transport (GOCART)[48] bulk aerosol scheme rather than the MOSAIC 4-bin aerosol scheme used in this study. Simulations over India comparing GOCART and MOSAIC 8-bin aerosol schemes have shown the MOSAIC scheme to better represent aerosol observations[49]. The MOSAIC 4-bin aerosol scheme is less computationally demanding relative to the 8-bin scheme, while performing well over India[49,50]. The model was evaluated against ECMWF re-analysis (ERA)[51] for boundary layer height, precipitation, wind speed, wind direction and temperature (Supplementary Fig. 8–12).

**Emissions inventory description**. Anthropogenic emissions were taken from the Emission Database for Global Atmospheric Research with Task Force on Hemispheric Transport of Air Pollution (EDGAR-HTAP) version 2.2 at 0.1 × 0.1° horizontal resolution[19]. EDGAR-HTAPv2.2 uses the Model Intercomparison Study for Asia Phase III (MIX), which is a mosaic Asian anthropogenic emission inventory[52]. For India, MIX used the Indian emission inventory provided by Argonne National Laboratory[53,54] for SO2, BC and OC for all sectors as well as $NO_x$ for power plants, and REAS2.1[55] for other species. Gaps in EDGAR-HTAPv2.2 were filled by the bottom-up global emission inventory EDGARv4.3. Emissions include SO2, $NO_x$, CO, NMVOC, NH3, BC and OC. Emissions are classified by source sector: aviation, shipping, power generation, industrial non-power, land transport, residential energy use and agriculture. Emissions from residential energy use categorised in EDGAR-HTAPv2.2 comprise small-scale combustion devices for heating, cooking, lighting and cooling in addition to supplementary engines for residential, commercial, agricultural, solid waste and wastewater treatment[19]. Residential energy use emissions of PM, BC and OC are qualitatively classified as highly uncertain within EDGAR-HTAPv2.2. Biomass burning emissions were taken from the Fire Inventory from NCAR (FINN) version

1.5[56]. Biogenic emissions were calculated online by the Model of Emissions of Gases and Aerosol from Nature (MEGAN)[57]. Dust emissions were calculated online through GOCART with Air Force Weather Agency (AFWA) modifications[48].

We calculate the contribution of specific emission sectors to $PM_{2.5}$ concentrations, through switching off emission sectors one at a time. The emission sectors investigated were agriculture (AGR), biomass burning (BBU), dust (DUS), power generation (ENE), industrial non-power (IND), residential energy use (RES) and land transport (TRA). All were annual simulations for 2014.

**Air quality evaluation**. Surface measurements of $PM_{2.5}$, O3, CO, NO2, SO2 were obtained from the Central Pollution Control Board (CPCB), Ministry of Environment and Forests, Government of India[1]. Details of the monitoring sites are given in Supplementary Table 4. The network of sites reporting $PM_{2.5}$ has expanded substantially in the last few years, with only four sites reporting in 2014 compared to 45 in 2016. India has strong seasonal variations in aerosol concentrations, but smaller interannual variability. $PM_{2.5}$ observations from 2016 were therefore selected to evaluate simulated annual and seasonal mean $PM_{2.5}$ from the model for 2014, as the order of magnitude increase in number of observation sites strengthens the evaluation statistics. Model evaluation for O3, CO, NO2 and SO2 are given in Supplementary Figs. 13 and 14.

**AOD evaluation**. Simulated AOD was evaluated against ground measurements (AERONET) and satellite (MODIS C6). Daily mean AOD data from 2014 were used from the ground-based AERONET sites in South Asia given in Supplementary Table 5. Level-2 (version 2) data were used, which are cloud-screened and quality assured with a low uncertainty of 0.01–0.02 at 500 nm. AOD was obtained at wavelengths between 340 and 1640 nm and was interpolated to obtain AOD at 550 nm. Simulated AOD was also evaluated against satellite AOD from the MODIS Aqua (MYD) satellite, using collection 6 (C6) level 2 (L2) AOD at 0.55 μm with the scientific data set 'optical depth land and ocean mean', which has a spatial resolution of 10 × 10 km (at nadir)[58]. The error in MODIS-derived AODs over land and ocean is ±0.05 + (0.15 × AOD) and ±0.03 + (0.05 × AOD), respectively. Daily means used model data only when the satellite retrieved observations, and used satellite data through a spatial-then-temporal approach with equal-day-weighting. Further information is discussed in Supplementary Methods.

**Meteorological evaluation**. Meteorological evaluation was undertaken using the European Centre for Medium-Range Weather Forecasts (ECMWF) global reanalysis products[51]. Daily means were downloaded for boundary layer height, precipitation, wind speed, wind direction and temperature. Estimates between 0° to 40° north and 60° to 100° east were acquired at 0.25° × 0.25° resolution, and seasonal means were determined for comparison with model output. The spatial variability in boundary layer height was generally well simulated apart from model overestimation during spring (Supplementary Fig. 8). Precipitation on land was reasonably estimated for winter and spring, though largely underestimated in summer and autumn (Supplementary Fig. 9). Seasonal variability in winds was well captured by the model (Supplementary Fig. 10). Temperature was generally well simulated by the model for all seasons (Supplementary Fig. 11). Supplementary Fig. 12 shows a scatter plot comparing annual mean simulated and reanalysis output for each meteorological variable.

**Population data**. The population count ($P$) data set at 0.25° × 0.25° resolution was obtained from the Gridded Population of the World, Version 4 (GPWv4), created by the Centre for International Earth Science Information Network (CIESIN) and accessed from the National Aeronautics and Space Administration (NASA) Socioeconomic Data and Applications Centre (SEDAC)[2]. The United Nations adjusted version was implemented for 2015 with a total population for India of 1.302 billion. Population age composition was taken from the GBD2015 population estimates for 2015[59]. The results from this study include rural and urban splits where urban areas are defined as having a population density of at least 400 persons km$^{-2}$, as used in previous studies[13]. Shapefiles were used at the state level within India from Spatial Data Repository, The Demographic and Health Surveys Program[60] and the GADM database of Global Administrative Areas version 2.8[61].

**Health impact estimation description**. Ambient $PM_{2.5}$ is associated with many health impacts, including acute lower respiratory infection (ALRI), ischaemic heart disease (IHD), cerebrovascular disease (CEV), chronic obstructive pulmonary disease (COPD) and lung cancer (LC)[4,62]. Disease burden was estimated from ALRI for early, late and post neonatal, and populations between 1 and 80 years upwards in 5 year groupings. Disease burden was estimated from IHD, CEV, COPD and LC for adults over 25 years old, split into 5-year age groups. We used the IER functions from the GBD2015 with age-specific modifiers for each disease to estimate the RR of premature mortality due to exposure to various $PM_{2.5}$ concentrations[3,4]. We used the parameter distributions of α, β and γ from the GBD2015 for 1000 simulations to derive the mean IER with 95% uncertainty intervals[3,4]. The IER functions have uniform theoretical minimum risk exposure levels (TMREL) for $PM_{2.5}$ between 2.4–5.9 μg m$^{-3}$. The outlined methods assume that the IERs are valid across the entire region.

Supplementary Fig. 5 shows calculated RR for the different diseases, highlighting the non-linear relationship between RR and $PM_{2.5}$ concentrations, particularly for $PM_{2.5}$ concentrations above 50 µg m$^{-3}$. Equation 1 expresses premature mortality ($M$) from disease endpoint ($j$) in grid cell ($i$) as a function of the population of the grid cell ($P$), the baseline mortality rate ($I$) and relative risk (RR) at the $PM_{2.5}$ concentration ($c$). Regional estimates were then calculated through summing all disease endpoints ($j$) over all grid cells ($i$), and split by state using shapefiles.

$$M_{i,j} = P_i I_j (RR_{j,c} - 1)/RR_{j,c}. \tag{1}$$

To be consistent with the GBD2015, we used country- and disease-specific baseline mortality rates from the GBD2015 study in 5-year groupings for both genders combined[63]. A sensitivity study was performed using state-specific baseline mortality rates[7] for India accounting for socioeconomic variations across the country through using gross domestic product (GDP) as a proxy applied to WHO statistics from 2011[64]. The sensitivity study applied the state-to-nation ratios from the state-specific baseline mortality rates to the GBD2015 baseline mortalities for COPD, IHD and CEV. Baseline mortality for LC did not exhibit any relation with GDP and they did not study ALRI, accordingly we directly used the GBD2015 value for these diseases. This was done for mean, upper and lower confidence intervals.

Years of life lost (YLL) are estimated following Eq. 2[65], where the number of deaths per disease and grid cell ($M_{i,j}$) is multiplied by the age-specific life expectancy (LE) remaining at the age of death from the standard reference life table from GBD2015[66].

$$YLL_{i,j} = M_{i,j} LE. \tag{2}$$

Country-specific life expectancy values[67] from the Government of India in 2014 were used in a sensitivity study to estimate 9,856,000 YLL (95UI: 4,763,000–12,549,000), 60% lower than when using the GBD2015 LE values. This study estimates health impacts from long-term exposure of whole populations to annual mean $PM_{2.5}$. This study does not account for indoor exposure to pollution, and the health impacts resulting from ambient $PM_{2.5}$ exposure therefore do not represent the total $PM_{2.5}$ related premature mortality burden. Household air pollution is a serious issue and there is a need to address this in conjunction with ambient air pollution in India[68].

Sector-specific mortality was calculated using two different methods: subtraction and attribution[23]. The subtraction method calculates the sector-specific premature mortality ($M_{SECTOR}$) as the difference between the premature mortality from all sources ($M_{ALL}$) and the premature mortality when one sector has been removed ($M_{SECTOR\_OFF}$) as in Eq. 3:

$$M_{SECTOR} = M_{ALL} - M_{SECTOR\_OFF}. \tag{3}$$

The attribution method first calculates the fractional sectoral reduction in $PM_{2.5}$ concentrations from removing an emission sector ($PM_{2.5\_SECTOR\_OFF}$) and then uses this fraction to scale the total premature mortality estimate (Eq. 4).

$$M_{SECTOR} = M_{ALL}(PM_{2.5\_ALL} - PM_{2.5\_SECTOR\_OFF})/PM_{2.5\_ALL}. \tag{4}$$

**Uncertainties**. We estimate an error in each term, then combine the fractional errors in quadrature (i.e. square root of the sum of squares). Uncertainty intervals at the 95% level (95UI) were determined reflecting the statistical uncertainty of the parameters in Eq. 1[69]. This includes the population data for India having an uncertainty range of ±2%[2]. The 95UI in annual mean $PM_{2.5}$ concentrations was estimated for each grid cell through assuming a Gaussian distribution and applying ±2 standard deviations from weekly $PM_{2.5}$ concentrations. The uncertainties in $PM_{2.5}$ are then applied to the derived uncertainties in the IER for the RR at both 5% and 95% confidence levels. The GBD2015[63] and state-specific[7] baseline mortality estimates have defined upper and lower uncertainty values. There are multiple other sources of uncertainty that are difficult to quantify. Emissions inventories for India have large uncertainties, especially across the IGP[70]. The model horizontal resolution of 30 km is unable to capture spatial variations at shorter scales. All fine particles are treated as equally toxic without regard to their source, shape and chemical composition.

**Code availability**. Code used in this study is available from the authors upon reasonable request.

**Data availability**. All health data created are openly available from the University of Leeds data archive at https://doi.org/10.5518/158. Other data are available from the authors upon reasonable request.

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

## Acknowledgements

L.C. acknowledges the support from the EPSRC CDT in Bioenergy (Grant No.EP/L014912/1). This work made use of the facilities of N8 High Performance Computing Centre of Excellence, provided and funded by the N8 consortium and EPSRC (Grant No. EP/K000225/1). The Centre is co-ordinated by the Universities of Leeds and Manchester. We acknowledge the use of the WRFotron scripts developed by Christoph Knote to automatise WRF-Chem runs with re-initialised meteorology. We acknowledge the use of WRF-Chem preprocessor tools mozbc, fire_emiss, anthro_emiss, bio_emiss provided by

NCAR, in addition to the post-processing script 'wrfout_to_cf.ncl' created by Mark Seefeldt at the University of Colorado at Boulder (http://foehn.colorado.edu/wrfout_to_cf/). We acknowledge Brent Holben, S. N. Tripathi, Panuganti C. S. Devara, Swagata Payra, Philippe Goloub, Gerrit de Leeuw, Rick Wagener, Laurie Gregory, Shubha Verma, Priya K. L. and their staff for establishing and maintaining the Aerosol Network (AERONET) sites used in this investigation. We acknowledge ECMWF re-analysis (ERA) for boundary layer height, precipitation, wind speed, wind direction and temperature (http://apps.ecmwf.int/datasets/).

## Author contributions

L.C., D.V.S., S.R.A. and C.K. designed the research. C.K. and L.C. setup the model. L.C. performed the model simulations, model evaluation, data analysis and wrote the manuscript. E.B. derived the IER. L.C., D.V.S., S.R.A. and C.K. evaluated the results. All authors commented on the manuscript.

## Additional information

**Competing interests:** The authors declare no competing financial interests.

