## [Peer Review File · Nature Communications]

Reviewers' comments:

Reviewer #1 (Remarks to the Author):

Review of Underestimated mortality from ambient air pollution in India

The authors estimate premature mortality from chronic exposure to ambient PM_{2.5} in India from all sources and from residential combustion emissions using an atmospheric model with higher spatial resolution than commonly used in many global modeling studies. The authors compare modeled ambient PM_{2.5} concentrations with surface observations. The novelty of this study is in estimating PM_{2.5} and resulting mortality from residential combustion at this high model resolution along with an extended comparison to surface observations in India. The high spatial resolution and agreement with surface observations of PM_{2.5} is used to justify the statement that past studies have underestimated mortality in India due to ambient exposure to PM_{2.5}. This study is of interest to those working on understanding and improving residential combustion emissions (cookstoves, heating,..) and communities estimating mortality PM_{2.5} from all sources (or other source-specific studies) using atmospheric models.

General comments:

One main comment/concern I have is with the framing of this study. The title and a fair amount of the main text discuss mortality from exposure to all-source PM_{2.5}. Estimating all-source PM_{2.5} exposure over India at high spatial resolutions is not entirely new in itself. Brauer et al. (2015) estimated PM_{2.5} exposures at 0.1x0.1 degree horizontal resolution for GBD2013, which was updated in Cohen et al. (2017) for GBD2015 (this last paper likely was published after this article was submitted, but should be referenced in a revision). These studies estimate PM_{2.5} at approximately 11x11 km (at midlatitudes) compared to the 30x30 km in this study. There is not a discussion that Cohen et al. (2017) and Brauer et al. (2015) are at a finer (or similar) spatial resolution as this study. Additionally, the relationship between spatial resolution and mortality has been explored elsewhere (the authors mention this in the introduction; I also think it is mentioned in some of the past studies shown in Figure 4). Finally, the GBD2015 estimate does not really underestimate mortalities in India compared to this study (actually, the GBD2015 estimate is entirely within the given uncertainty range reported in this study). Could the conclusions of this section be drawn from the Cohen et al. (2017) study?

The high resolution estimates of residential combustion are new and interesting (and the authors state this is the novel aspect). The authors estimate mortality from residential emissions by scaling the total mortalities by the fraction of PM_{2.5} from residential emissions and by subtracting the mortalities if there were no residential emissions (approximating an intervention scenario). This is a useful update as past estimates used older concentration-response functions and PM_{2.5} estimates and coarser model resolutions. The scaling method is used to compare to past studies. In terms of the scaling method, I am curious if the fraction of PM_{2.5} from residential sources has also changed with spatial resolution (in terms of comparing to other studies). For instance, are the fewer mortalities in Lelieveld et al. (2015) due to fewer all-source mortalities but a similar fraction of PM_{2.5} from residential

sources or fewer all-source mortalities as well as a smaller fraction of PM2.5 from residential sources? Also looking at Figure 4, the Chafe et al. (2014) paper focuses mainly on outdoor mortalities to cookstove emissions (and splits apart this source from other residential sources). Is the number used in Figure 4 for Chafe et al. (2014) from just cookstove emissions or all residential emissions?

The comparison to past studies is interesting, but it is unclear to me what exact conclusion should be drawn from this. The authors test the sensitivity to changing spatial resolution and population estimates in order to compare to different mortality estimates. However, there are a number of other datasets and methods that do not appear to be controlled for, such as baseline mortality and (importantly) concentration-response function. I think this needs to be discussed more explicitly. For instance, is the lower mortality estimate in Butt et al. (2016) from model resolution or a concentration-response function with a substantially different mathematical form as used in this study?

Specific comments:

Figure 4 shows that Ghude et al. (2016) has a PM2.5 resolution of 0.36 degrees. Why is the mortality estimate in this study lower than studies with an even coarser model resolution?

The Brauer et al. (2015) 0.1x0.1 degree PM2.5 grid is used in the GBD2013 estimate. It would be interesting to know why this estimate produces a lower mortality estimate over India despite being at a similar resolution as GBD2015 and this study. Figure 4 indicates that PM2.5 in GBD2013 is at a 1x1 degree grid. Was the Brauer et al. (2015) estimate re-gridded during the mortality calculation in GBD2013?

I think the title of the paper is misleading. The title references "air pollution", but this study only considers ambient PM2.5 (and not things like ozone).

I think the main text should include some discussion of what spatial resolution this study uses compared to what is common in other studies. Simply referring to "high resolution" is not entirely sufficient, as 30 km may itself be too coarse when thinking on the scale of villages in India. The actual model resolution is not shown until Figure 4 and the supplemental sections.

Why is the uncertainty range from all source mortality in this study so much larger than in GBD2015?

Additionally, there is no uncertainty range in the mortalities from residential combustion PM2.5 (unless I missed it). Do you assume the same relative uncertainty?

Did you use the age-specific modifiers for the concentration-response functions for IHD and CEV (and the corresponding baseline mortality estimates)?

Lines 26-28: Premature mortality isn't exactly exacerbated by high baseline mortality rates, this just comes in from attributing some number of total deaths (from all risk factors and from some disease) to PM exposure. This just needs to be re-worded. It sounds like you are

saying mortality is higher because mortality rates are higher.

Line 34: "Indian aerosol". I'm not sure if this is used elsewhere in literature, but it sounds a little odd to me. Maybe just "aerosol concentrations over India". The aerosol itself is not Indian or American.

References:

Ambient Air Pollution Exposure Estimation for the Global Burden of Disease 2013
Michael Brauer, Greg Freedman, Joseph Frostad, Aaron van Donkelaar, Randall V. Martin, Frank Dentener, Rita van Dingenen, Kara Estep, Heresh Amini, Joshua S. Apte, Kalpana Balakrishnan, Lars Barregard, David Broday, Valery Feigin, Santu Ghosh, Philip K. Hopke, Luke D. Knibbs, Yoshihiro Kokubo, Yang Liu, Stefan Ma, Lidia Morawska, José Luis Texcalac Sangrador, Gavin Shaddick, H. Ross Anderson, Theo Vos, Mohammad H. Forouzanfar, Richard T. Burnett, and Aaron Cohen
Environmental Science & Technology 2016 50 (1), 79-88
DOI: 10.1021/acs.est.5b03709

Cohen, A. J. et al. (2017), Estimates and 25-year trends of the global burden of disease attributable to ambient air pollution: an analysis of data from the Global Burden of Diseases Study 2015, Lancet, doi:10.1016/S0140-6736(17)30505-6.

Reviewer #2 (Remarks to the Author):

Air pollution is important for health, and India is a region with very large health impacts that has not been the focus of sufficient attention. So the context of this paper is very important. The authors use a fine resolution model of air pollution over India to 1) estimate the health effects of exposure to PM_{2.5}, and 2) find the contribution of residential emissions to this total burden. The first goal is not new – the authors state that (l. 15-16) "previous studies have underestimated this health burden ... by a factor of two". But in fact only a few papers are a factor of two lower than their current estimate, and in fact the most recent GBD results are within a few percent of the current paper. The GBD uses methods that are better than the current paper, as they make use of satellite observations, monitored concentrations, and a model, and combine these sources of information together statistically to get concentrations at even finer resolution than the current study – in contrast, the present paper uses only a model, albeit at fine resolution. The authors also reference a previous fine-resolution model for India (reference 19), so the claim of novelty on resolution is not supported. The second goal is new in focusing on India, but previous studies have done this attribution globally using similar methods, but for more source categories. Again the novelty is not clear.

Overall, I am persuaded that the present paper estimated PM_{2.5}-related mortality in a way that is supported by the current literature, and therefore can make a good contribution to the literature. But the paper is not very novel, nor do results differ sufficiently that publication in a high-impact journal would be warranted. Instead I would recommend that

the authors pursue publication in a disciplinary journal (EHP, ES&T, ACP, or others).

More specific comments

The authors focus on PM_{2.5}, which kills more people than ozone, but it would not have been difficult for them to also include ozone mortality.

The authors simulate 2014 but then compare with observations from 2016. I can guess why they did this, but the mismatch is a weakness of the paper. The authors should justify their choice, and discuss the uncertainties it brings.

I. 11 – The abstract mentions the importance of new 2016 observations, and it is true that ground observations were limiting the quality of previous studies, but it is not clear that those observations indicate higher PM_{2.5} than was previously believed, nor that they led to higher estimates in this study. In fact, it seems that the new observations were only used to evaluate the model, as the model was not bias-corrected. The novelty of the new observations was not then explained in the main paper.

I. 13 – The title emphasizes “underestimated mortality” but here the authors state that their purpose is to estimate “the impacts of residential combustion”. Please be more clear on your purpose (which it seems are two).

I. 32 – It is true that India has high baseline mortality rates, but it does not have high baseline rates for the causes of disease that are affected by PM_{2.5}.

I. 65-76 – The authors give a long discussion of different methods, which I don’t think is necessary. But then they don’t clarify which method they use!

I. 133 – The authors mention how results would differ had they used state-specific mortality rates. I don’t see where they mention what mortality rates they do use. And if they know that baseline rates are higher in some states, then why couldn’t state-specific rates be used?

Figure 3 – The authors should discuss why the two panels are so different.

Modeling methods (p. 11) – the modeling seems to be done well in general. However, the MOZART-4 chemical mechanism is really created for global applications. The authors would have been better to use a regional mechanism with more detail in fast-reacting chemistry. This choice weakens the authors’ arguments that the fine resolution is a strength of this study.

p. 15 – This is a discussion of uncertainties, but I don’t see that uncertainties are presented for the main results – in the abstract or Figure 4. If uncertainties are estimated, how is uncertainty from several different factors propagated to give an overall uncertainty? “2 standard deviations” – I am not clear on what population of values the 2 standard deviations is chosen from.

Conibear et al.

We are very grateful for the fair and helpful comments from both reviewers. We have responded to all comments from both reviewers. We think that these important comments help us demonstrate convincingly that residential energy use dominates exposure to ambient PM_{2.5} and is the leading cause of mortality related to PM_{2.5} exposure across India.

Reviewers' comments are in *italics*.

Our responses are in **bold**. Additions and changes to the manuscript are identified with line numbers.

Reviewer #1 (Remarks to the Author):

Review of Underestimated mortality from ambient air pollution in India

The authors estimate premature mortality from chronic exposure to ambient PM_{2.5} in India from all sources and from residential combustion emissions using an atmospheric model with higher spatial resolution than commonly used in many global modeling studies. The authors compare modeled ambient PM_{2.5} concentrations with surface observations. The novelty of this study is in estimating PM_{2.5} and resulting mortality from residential combustion at this high model resolution along with an extended comparison to surface observations in India. The high spatial resolution and agreement with surface observations of PM_{2.5} is used to justify the statement that past studies have underestimated mortality in India due to ambient exposure to PM_{2.5}. This study is of interest to those working on understanding and improving residential combustion emissions (cookstoves, heating,..) and communities estimating mortality PM_{2.5} from all sources (or other source-specific studies) using atmospheric models.

General comments:

One main comment/concern I have is with the framing of this study. The title and a fair amount of the main text discuss mortality from exposure to all-source PM_{2.5}. Estimating all-source PM_{2.5} exposure over India at high spatial resolutions is not entirely new in itself. Brauer et al. (2015) estimated PM_{2.5} exposures at 0.1x0.1 degree horizontal resolution for GBD2013, which was updated in Cohen et al. (2017) for GBD2015 (this last paper likely was published after this article was submitted, but

should be referenced in a revision). These studies estimate PM_{2.5} at approximately 11x11 km (at midlatitudes) compared to the 30x30 km in this study. There is not a discussion that Cohen et al. (2017) and Brauer et al. (2015) are at a finer (or similar) spatial resolution as this study. Additionally, the relationship between spatial resolution and mortality has been explored elsewhere (the authors mention this in the introduction; I also think it is mentioned in some of the past studies shown in Figure 4). Finally, the GBD2015 estimate does not really underestimate mortalities in India compared to this study (actually, the GBD2015 estimate is entirely within the given uncertainty range reported in this study). Could the conclusions of this section be drawn from the Cohen et al. (2017) study?

Thank you for this helpful point. We agree and we have reframed the paper accordingly to focus on the contribution from residential energy use emissions, rather than all source PM_{2.5}. Building on the novel aspect of the contribution from residential energy use emissions, we have included the results from six additional model simulations studying other source categories (as outlined above). Results from these simulations further confirm that residential energy use emissions are the dominant contributor to PM_{2.5} and associated mortality across India. See Table 1, Table 2 and Figure 3b.

As discussed above we have reduced our discussion on all-source PM_{2.5}. We have added discussion of Cohen et al. (2017) and Brauer et al. (2016) to our comparison to previous studies. We agree with the reviewer that spatial resolution is not the main factor causing differences between previous studies and we agree that the issue of resolution has been discussed in previous studies. We therefore have reduced the discussion of spatial resolution and we highlight that resolution was not the driving factor in the differences in mortality estimates. The total premature mortality in India from the Cohen et al. (2017) is within 5% of our estimated mortality. We highlight the agreement between our study and Cohen et al. (2017) and we are careful in our wording to state that we think the Cohen et al. (2017) result is not an underestimate. We estimate annual mean PM_{2.5} concentrations of up to 150 µg m⁻³ across the Indo-Gangetic Plain (IGP) (Figure 1a), consistent with observations and similar to that simulated by Cohen et al. (2017). We think it is important to point out that many other previous

studies, including Brauer et al. (2016), estimated maximum annual mean $PM_{2.5}$ concentrations across the IGP to be between $50-80 \mu g m^{-3}$, up to a factor of 2 lower than observations. We note that evaluation of these results was limited by lack of Indian $PM_{2.5}$ observations available at that time. The substantially lower estimates of $PM_{2.5}$ concentrations in these studies directly reduces premature mortality estimates from exposure to ambient $PM_{2.5}$ over India, outweighing the variation due to differences in exposure-response relationship, baseline mortality, population and resolution (as shown in sensitivity studies and explained below in response to your comment). We have clarified these points in our manuscript, see lines 137-158 and Figure 4.

The high resolution estimates of residential combustion are new and interesting (and the authors state this is the novel aspect). The authors estimate mortality from residential emissions by scaling the total mortalities by the fraction of $PM_{2.5}$ from residential emissions and by subtracting the mortalities if there were no residential emissions (approximating an intervention scenario). This is a useful update as past estimates used older concentration-response functions and $PM_{2.5}$ estimates and coarser model resolutions. The scaling method is used to compare to past studies. In terms of the scaling method, I am curious if the fraction of $PM_{2.5}$ from residential sources has also changed with spatial resolution (in terms of comparing to other studies). For instance, are the fewer mortalities in Lelieveld et al. (2015) due to fewer all-source mortalities but a similar fraction of $PM_{2.5}$ from residential sources or fewer all-source mortalities as well as a smaller fraction of $PM_{2.5}$ from residential sources? Also looking at Figure 4, the Chafe et al. (2014) paper focuses mainly on outdoor mortalities to cookstove emissions (and splits apart this source from other residential sources). Is the number used in Figure 4 for Chafe et al. (2014) from just cookstove emissions or all residential emissions?

Lelieveld et al. (2015) estimated the contribution of emissions from residential energy use to total ambient $PM_{2.5}$ concentrations (50%) and multiplied the total mortality associated with ambient $PM_{2.5}$ over India (645,000 per year) by this fraction, giving 322,000 premature mortalities per year from residential emissions. We estimate a similar fraction (52%) of residential energy use emissions to total ambient $PM_{2.5}$ concentrations, confirming that it did

not change much with spatial resolution. When we apply the same scaling method as Lelieveld et al. (2015) we get 541,000 premature mortalities per year (see Table 2), due to our higher total mortality (1,038,000 lives per year). However, as we show in the paper this method overestimates source specific mortality in regions with high $PM_{2.5}$ concentrations. When we remove emissions from residential energy use and recalculate premature mortality, we calculate 268,000 premature mortalities per year. In summary, our overall result for residential emissions is similar to Lelieveld et al. (2015), but due to a combination of their lower estimate of total mortality combined with their higher estimate of the reduction in mortality that would occur when residential emissions are removed. We feel that our new discussion more clearly makes this point, without specifically comparing individual studies. See Table 2 and lines 160-187. Also, thank you for pointing out the mistake in Figure 4, the estimate from Chafe et al. (2014) is just from cooking emissions.

The comparison to past studies is interesting, but it is unclear to me what exact conclusion should be drawn from this. The authors test the sensitivity to changing spatial resolution and population estimates in order to compare to different mortality estimates. However, there are a number of other datasets and methods that do not appear to be controlled for, such as baseline mortality and (importantly) concentration-response function. I think this needs to be discussed more explicitly. For instance, is the lower mortality estimate in Butt et al. (2016) from model resolution or a concentration-response function with a substantially different mathematical form as used in this study?

We have now expanded our analysis to compare baseline mortality and exposure-response functions. Specifically, we have added a sensitivity calculation using the GBD2010 integrated-exposure response (IER) function and the GBD2015 baseline mortality estimates for 2010. Using the RR from the GBD2010 IER, increased our mortality estimates by 22%. Many previous studies we identify as underestimating the premature mortality due to ambient $PM_{2.5}$ exposure over India used the GBD2010 IER (i.e. Lelieveld et al. (2015), Silva et al. (2016), Ghude et al. (2016), Apte et al. (2015), Chowdhury and Dey (2016), Giannadaki et al. (2016)). As this

sensitivity study increased our mortality estimates, the underestimation of these previous studies cannot therefore be attributed to the different exposure-response function. Our new analysis confirms more clearly that a likely large reason for lower premature mortality estimates in many previous studies is due to lower simulated PM_{2.5} concentrations in previous studies.

We now expand our discussion (see lines 150-153) and have expanded Figure 4 to include this new analysis. We agree with the referee that this is not the most important conclusion of our study, and potentially these conclusions could be drawn from Cohen et al. (2017). However, we think that it is useful to highlight this complex combination of different factors and in particular the lower PM_{2.5} concentrations in many studies published prior to 2016 and availability of observations. We feel this really highlights the crucial importance of new PM_{2.5} observations across areas with sparse observations.

The lower mortality estimate from Butt et al. (2016) is a combination of using the log-linear exposure-response model of Ostro (2004), with substantially lower PM_{2.5} concentrations than this study, the GBD2015 / Cohen et al. (2017) and observations.

Specific comments:

Figure 4 shows that Ghude et al. (2016) has a PM_{2.5} resolution of 0.36 degrees. Why is the mortality estimate in this study lower than studies with an even coarser model resolution?

Ghude et al. (2016) estimate similar maximum annual mean PM_{2.5} concentrations to this study, GBD2015 / Cohen et al. (2017) and observations. They use the GBD2010 exposure-response functions, population data from the 2011 Indian Census, and baseline mortalities from WHO 2012. Ghude et al. (2016) estimated a similar mortality burden to GBD2013 / Brauer et al. (2016), while Ghude et al. (2016) having higher PM_{2.5} concentrations and a higher RR response to PM_{2.5}.

The Brauer et al. (2015) 0.1x0.1 degree PM_{2.5} grid is used in the GBD2013 estimate. It would be interesting to know why this estimate produces a lower mortality estimate over India despite being at

a similar resolution as GBD2015 and this study. Figure 4 indicates that PM_{2.5} in GBD2013 is at a 1x1 degree grid. Was the Brauer et al. (2015) estimate re-gridded during the mortality calculation in GBD2013?

Sorry, the 1x1 degree grid caption in figure 4 was a mistake. We did not re-grid the Brauer et al. (2016) / GBD2013 data.

Brauer et al. (2016) estimated maximum annual mean PM_{2.5} concentrations across the IGP to be between 50-80 µg m⁻³, up to a factor of 2 smaller than observations, our study and the GBD2015 / Cohen et al. (2017). This leads to the lower mortality estimates in the GBD2013 of 587,000 for India. Furthermore, the exposure-response function used within GBD2013 used less epidemiological data and consequently had a different form to the one used within GBD2015, and estimated lower RR than for GBD2015 for the two critical diseases (COPD and IHD) contributing to the disease burden in India. We did calculate the premature mortality using the GBD2013 IER, and this reduced our mortality estimates by 31%, to 719,000 premature mortalities per year. We therefore think a key reason for the lower mortality estimate is the lower simulated PM_{2.5} concentrations.

I think the title of the paper is misleading. The title references “air pollution”, but this study only considers ambient PM_{2.5} (and not things like ozone).

Thank you for this useful suggestion. We agree and have changed the title accordingly.

I think the main text should include some discussion of what spatial resolution this study uses compared to what is common in other studies. Simply referring to “high resolution” is not entirely sufficient, as 30 km may itself be too coarse when thinking on the scale of villages in India. The actual model resolution is not shown until Figure 4 and the supplemental sections.

We now mention the model resolution earlier in the text (see lines 60-62), and discussed the different resolutions used in many previous studies (see lines 140-148). We note Figure 1 shows simulated PM_{2.5} concentrations at the model resolution. We agree that even 30 km is too coarse to resolve the scale of villages. However, as mentioned earlier, we find the differences between

previous studies are being driven less by differences in resolution and more by lower simulated PM_{2.5} concentrations and older exposure-response functions.

Why is the uncertainty range from all source mortality in this study so much larger than in GBD2015?

Differences in uncertainty ranges likely arise due to the uncertainty propagating from simulated PM_{2.5} in this study is defined as ± 2 standard deviations for each grid cell, rather than the sophisticated statistical approach the GBD2015 / Cohen et al. (2017) / Shaddick et al. (2017) uses. We have now improved our uncertainty estimate by combining fractional errors in quadrature (see lines 312-318), which has reduced the uncertainty range and is now more in lines with GBD2015.

Additionally, there is no uncertainty range in the mortalities from residential combustion PM_{2.5} (unless I missed it). Do you assume the same relative uncertainty?

Sorry we have added the uncertainty ranges to the text and they are derived using the same methodology as for all sources. See lines 22, 118-119, 167, 197 and Table 2.

Did you use the age-specific modifiers for the concentration-response functions for IHD and CEV (and the corresponding baseline mortality estimates)?

Yes, we have clarified this in the paper now (see lines 298-301).

Lines 26-28: Premature mortality isn't exactly exacerbated by high baseline mortality rates, this just comes in from attributing some number of total deaths (from all risk factors and from some disease) to PM exposure. This just needs to be re-worded. It sounds like you are saying mortality is higher because mortality rates are higher.

We have adjusted text to just focus on the baseline mortality rates we use, and why we didn't use the state specific ones (see lines 298-301).

Line 34: “Indian aerosol”. I’m not sure if this is used elsewhere in literature, but it sounds a little odd to me. Maybe just “aerosol concentrations over India”. The aerosol itself is not Indian or American.

This has been removed through adjusting text to account for other comments (see lines 29-31).

References:

Ambient Air Pollution Exposure Estimation for the Global Burden of Disease 2013

Michael Brauer, Greg Freedman, Joseph Frostad, Aaron van Donkelaar, Randall V. Martin, Frank Dentener, Rita van Dingenen, Kara Estep, Heresh Amini, Joshua S. Apte, Kalpana Balakrishnan, Lars Barregard, David Broday, Valery Feigin, Santu Ghosh, Philip K. Hopke, Luke D. Knibbs, Yoshihiro Kokubo, Yang Liu, Stefan Ma, Lidia Morawska, José Luis Texcalac Sangrador, Gavin Shaddick, H. Ross Anderson, Theo Vos, Mohammad H. Forouzanfar, Richard T. Burnett, and Aaron Cohen

Environmental Science & Technology 2016 50 (1), 79-88

DOI: 10.1021/acs.est.5b03709

Cohen, A. J. et al. (2017), Estimates and 25-year trends of the global burden of disease attributable to ambient air pollution: an analysis of data from the Global Burden of Diseases Study 2015, Lancet, doi:10.1016/S0140-6736(17)30505-6.

Reviewer #2 (Remarks to the Author):

Air pollution is important for health, and India is a region with very large health impacts that has not been the focus of sufficient attention. So the context of this paper is very important. The authors use a fine resolution model of air pollution over India to 1) estimate the health effects of exposure to PM2.5, and 2) find the contribution of residential emissions to this total burden. The first goal is not new – the authors state that (l. 15-16) “previous studies have underestimated this health burden ... by a factor of two”. But in fact only a few papers are a factor of two lower than their current estimate, and in fact the most recent GBD results are within a few percent of the current paper. The GBD uses methods that are better than the current paper, as they make use of satellite observations, monitored

concentrations, and a model, and combine these sources of information together statistically to get concentrations at even finer resolution than the current study – in contrast, the present paper uses only a model, albeit at fine resolution. The authors also reference a previous fine-resolution model for India (reference 19), so the claim of novelty on resolution is not supported. The second goal is new in focusing on India, but previous studies have done this attribution globally using similar methods, but for more source categories. Again the novelty is not clear.

Overall, I am persuaded that the present paper estimated PM_{2.5}-related mortality in a way that is supported by the current literature, and therefore can make a good contribution to the literature. But the paper is not very novel, nor do results differ sufficiently that publication in a high-impact journal would be warranted. Instead I would recommend that the authors pursue publication in a disciplinary journal (EHP, ES&T, ACP, or others).

Thank you for your comments. We agree about your point on focusing on all source mortality is not novel and we have adjusted accordingly to focus on the novel aspect of source contributions at high resolution, evaluated against extensive new observations. We have added the contribution of six additional source categories, strengthening this novel aspect and the main conclusion that residential energy use is the dominant contributor. We agree that the GBD uses a sophisticated method to combine model with observations and satellite data to estimate PM_{2.5} concentrations. However, using this method means that the GBD cannot calculate the contribution of different emission sectors. This is particularly important given the issues we show around simply scaling PM_{2.5} or mortality.

With the additions and corrections suggested by the editor and both reviewers, our paper now demonstrates convincingly that residential energy use is the dominant contributor to ambient PM_{2.5} and related mortality across India. We hope that this addresses your comment on the novelty of our paper. See Table 1, Table 2 and Figure 3b.

We apologise for our lack of clarity in the text regarding saying that all previous studies have underestimated, and we have reworded to state that our total PM_{2.5} mortality estimates agree

within 5% of Cohen et al. (2017). We clarify that other previous studies likely underestimated due lower PM_{2.5} concentrations being up to a factor of 2 less than observations, sometimes coupled to different exposure-response functions giving lower RR and older population datasets. We have updated our discussion on this point (see lines 137-158).

We agree that the GBD uses a sophisticated combination of model, satellite and observations at high resolution. However, this sophisticated combination of model, satellite and observations makes it difficult for the GBD to simulate the contribution of different emission sectors to PM_{2.5}. Furthermore, we note that maximum PM_{2.5} concentrations across the Indian IGP in Brauer et al. (2016) are up to half of that of either Cohen et al. (2017) or our study. Studies before 2016 were limited when evaluating their PM_{2.5} concentrations over India by a lack of extensive ground observations of PM_{2.5} across India, and it is this underestimation of PM_{2.5} concentrations which is likely a key reason why the associated mortality estimates for India are substantially lower in GBD2013 than for GBD2015, in addition to the different IER used in each GBD assessment.

We think that our study focusing on the source contributions to the burden of disease attributable to ambient PM_{2.5} exposure in India, simulated with a high-resolution model and evaluated against extensive new ground observations, is both novel and an important addition to addressing this serious issue.

More specific comments

The authors focus on PM_{2.5}, which kills more people than ozone, but it would not have been difficult for them to also include ozone mortality.

We have decided to focus on ambient PM_{2.5} in this study, due to the dominant role of PM_{2.5} in ambient air pollution associated mortality. Cohen et al. (2017) for the GBD2015 estimate annual mortality for India in 2015 from ambient PM_{2.5} (1,090,390) to be an order of magnitude larger than from ambient O₃ (107,770).

The authors simulate 2014 but then compare with observations from 2016. I can guess why they did this, but the mismatch is a weakness of the paper. The authors should justify their choice, and discuss the uncertainties it brings.

The justification has been moved for clarity (see lines 260-264 and explained below). The network of sites reporting PM_{2.5} has expanded substantially in the last few years, with only 4 sites reporting in 2014 compared to 45 in 2016. We were unable to run our model specifically for 2016 due to unavailability of all the driving datasets required. India has strong seasonal variations in aerosol concentrations, but smaller interannual variability. For these reasons, PM_{2.5} data from 2016 was selected to evaluate the model for 2014. We restricted our evaluation to annual and seasonal mean PM_{2.5}. We note that we were able to evaluate the model against AERONET using observations of AOD from 2014 and this shows a similar level of agreement. This further confirms limited interannual variability and confirms our approach is appropriate given the limitations in data availability. We have added this evaluation against AERONET to the main paper (see lines 74-75).

l. 11 – The abstract mentions the importance of new 2016 observations, and it is true that ground observations were limiting the quality of previous studies, but it is not clear that those observations indicate higher PM_{2.5} than was previously believed, nor that they led to higher estimates in this study. In fact, it seems that the new observations were only used to evaluate the model, as the model was not bias-corrected. The novelty of the new observations was not then explained in the main paper.

The new observations across India show maximum annual mean PM_{2.5} concentrations of approximately 150 µg m⁻³. The maximum annual mean PM_{2.5} concentrations across India simulated in this study well matches the observations, as does Cohen et al. (2017). Seven of the previous studies, including Brauer et al. (2016), estimate maximum annual mean PM_{2.5} concentrations across the Indian IGP between 50-80 µg m⁻³. The relative risks for all diseases at

50 $\mu\text{g m}^{-3}$ are substantially lower than the relative risks at 150 $\mu\text{g m}^{-3}$, and this directly lowers the mortality estimates.

The referee is correct that we used the observations for model evaluation and did not bias-correct the model. The important point is that these are new observations that have only just become available and allow a detailed evaluation of $\text{PM}_{2.5}$ concentrations across India for the first time. This allows us to demonstrate that the WRF-Chem model realistically reproduces $\text{PM}_{2.5}$ concentrations over India. Previous studies were unable to evaluate their model in the same way, meaning there was little constraint on simulated concentrations. These studies were therefore not aware that they were likely seriously underestimating $\text{PM}_{2.5}$ concentrations across India. We clarify this in the main paper and identifying the simulated $\text{PM}_{2.5}$ concentrations previous studies simulated (see lines 154-156). As we state elsewhere, this demonstrates the crucial importance of new $\text{PM}_{2.5}$ observations in regions with sparse or non-existent observations.

l. 13 – The title emphasizes “underestimated mortality” but here the authors state that their purpose is to estimate “the impacts of residential combustion”. Please be more clear on your purpose (which it seems are two).

Sorry for this, we have corrected the title and the relevant parts of paper. See abstract, introduction, results and conclusion.

l. 32 – It is true that India has high baseline mortality rates, but it does not have high baseline rates for the causes of disease that are affected by $\text{PM}_{2.5}$.

We have adjusted the text to just focus on the baseline mortality rates we use, and why we didn't use the state specific ones (see lines 298-301).

l. 65-76 – The authors give a long discussion of different, methods, which I don't think is necessary. But then they don't clarify which method they use!

We have reduced the discussion of the different methods (see lines 47-56). We have clarified which method we use (see lines 62-65, 253-256). We show that highlighting the different methods is important when estimating the contribution of different pollution sources to mortality from PM_{2.5} exposure due to the substantially different results they produce (see Table 2 and lines 160-187).

l. 133 – The authors mention how results would differ had they used state-specific mortality rates. I don't see where they mention what mortality rates they do use. And if they know that baseline rates are higher in some states, then why couldn't state-specific rates be used?

We mention which baseline mortality rates we use and why in lines 298-301. The reasoning we use the baseline mortality rates from GBD2015 over the state-specific baseline mortality rates is that the state-specific ones do not vary with age, which is critical especially when estimating uncertainty intervals. Furthermore, the IER is age-dependent for IHD and CEV so we need age-specific baseline mortality rates.

Figure 3 – The authors should discuss why the two panels are so different.

Added to text (see lines 103-112). Residential energy use emissions contribute a larger fraction of anthropogenic PM_{2.5} emissions than to ambient PM_{2.5} concentrations, due to non-anthropogenic sources of PM_{2.5}, including dust and biomass burning and to the contribution of aerosol precursors (e.g. SO₂, NO_x).

Modeling methods (p. 11) – the modeling seems to be done well in general. However, the MOZART-4 chemical mechanism is really created for global applications. The authors would have been better to use a regional mechanism with more detail in fast-reacting chemistry. This choice weakens the authors' arguments that the fine resolution is a strength of this study.

Sorry, we were unclear in our description. In this study, we used the MOZART mechanism as implemented in WRF-Chem at the time when we conducted the simulations. While this mechanism is commonly referred to as "MOZART-4" Emmons et al. (2010), its implementation

in WRF-Chem has received multiple updates regarding fast photochemistry. These updates include (described in Knote et al. (2014) if not mentioned otherwise):

- Updated treatment of photochemistry of isoprene based on the Master Chemical Mechanism (University of Leeds)
- Explicit monoterpene species instead of lumped C₁₀H₁₆ species, Hodzic et al. (in prep.)
- Explicit treatment of C-5 unsaturated hydroxycarbonyl chemistry
- Separation of previously lumped aromatics species named “TOLUENE” into benzene, toluene and xylenes
- Chemistry of C₂H₂ (ethyne) and MBO (2-methyl-3-buten-2-ol)

With these modifications, the MOZART mechanism transitions from a mechanism focused on global model, with coarse grid chemistry to a mechanism that is on-par with other mechanisms considered standard for regional model gas-phase chemistry (e.g. SAPRC99, RACM, CB05). Its performance has been documented in Knote et al. (2015) in a box-model comparison, and within WRF-Chem in Knote et al. (2014) and the analyses made during the AQMEII phase 2 3-D model intercomparison (Campbell et al. (2015), Im et al. (2015), Wang et al. (2015)). In these studies, the same WRF-Chem setup (with respect to gas-phase chemistry) had been used that we employed in the manuscript under review here.

- Knote et al., (2014): <http://www.atmos-chem-phys.net/14/6213/2014/>
- Knote et al. (2015): <http://www.sciencedirect.com/science/article/pii/S1352231014009388>
- Hodzic et al., (in prep): Hodzic, A., Gochis D., Cui, Y., et al.: Meteorological conditions, emissions and transport of anthropogenic pollutants over the Central Rocky Mountains during the 2011 BEACHON-RoMBAS field study, in preparation.
- Campbell et al.
(2015): <http://www.sciencedirect.com/science/article/pii/S1352231014009741>
- Im et al. (2015): <http://www.sciencedirect.com/science/article/pii/S1352231014007353>
- Wang et al. (2015): <http://www.sciencedirect.com/science/article/pii/S1352231014005743>

Consequently, we have changed lines 210-212 to read: “Gas-phase chemical reactions are calculated using the chemical mechanism MOZART- 4, with several updates to photochemistry of aromatics, biogenic hydrocarbons and other species relevant to regional air quality.”

p. 15 – This is a discussion of uncertainties, but I don’t see that uncertainties are presented for the main results – in the abstract or Figure 4. If uncertainties are estimated, how is uncertainty from several different factors propagated to give an overall uncertainty? “2 standard deviations” – I am not clear on what population of values the 2 standard deviations is chosen from.

Apologies for this, the uncertainties have been added. See lines 22, 118-119, 167, 197 and Table 2. We estimate an error in each term then combine fractional errors in quadrature. The 2 standard deviations are applied to annual mean PM_{2.5} concentrations per grid cell, and the text has been edited for clarity. The uncertainties in PM_{2.5} are then applied to the derived uncertainties in the IER for the RR at both 5% and 95% confidence levels.

Additional improvements to paper

1. Updated to GBD2015 relative risks (from previously used GBD2010).
 - a. Method: lines 279-287, Figure 5
 - b. Mortality estimates: lines 22, 118-119, 167, 197, Table 2, Figures 3, 4, Supplementary Table 7, 8
2. Downloaded hourly ground measurement data for PM_{2.5}, NO₂, O₃, SO₂ and CO (previously was daily data) for all sites plus 10 additional. Evaluation against hourly data is similar to the previous daily observations (Previous: NMB = -0.10, slope = 0.76. Now: NMB = 0.05, slope = 0.83).
 - a. Method: lines 260-264
 - b. PM_{2.5} evaluation: lines 70-80, Figures 1a, 1b (also removed error bars for clarity), Supplementary Figures 1, 12, 13, 14, Supplementary Table 3, 5, Supplementary Methods: lines 142-153, 178-183
3. Improved AOD evaluation to only include model values at the daily overpass of satellite when satellite observations are available (i.e. clear sky).
 - a. AOD evaluation: lines 74-75, Supplementary Figure 9, 10, 11, Supplementary Table 5, Supplementary Methods: lines 165-177, 184-192
4. Misc.
 - a. Changed reference for population data and reference life table. Lines 274-275 and references 2 and 57
 - b. Author contributions (added dots for initials)
 - c. Format of both mortality and YLL equations. Lines 297, 305
 - d. Changed reference for GBD studies (references 3, 11, 13)

Reviewers' comments:

Reviewer #1 (Remarks to the Author):

Review of Emissions from residential energy use dominate exposure to ambient fine particulate matter in India

The authors estimate premature mortality in India due to exposure to PM_{2.5} from all sources and source-specific mortality rates. The authors include a comparison to surface observations and AOD in India. The revised version of this manuscript focuses on the large contribution of emissions from the residential energy sector to PM_{2.5} and associated premature mortality. The novel aspect of this work is sector-specific mortality estimates over India at a finer model resolution than what is used in typical global models.

Overall, the authors responded to all of my comments and the framing of the revised manuscript is more in line and supported by the results of the study.

I have a few additional concerns that that authors should address before I can recommend publication.

Main comments

1. In lines 50-52, the authors state that scaling the mortalities by the contribution of an emission sector to total ambient PM_{2.5} concentration "assumes a linear relationship between PM_{2.5} exposure and premature mortality"; however, I don't really agree with this. The relative risk (RR) is calculated with the nonlinear concentration-response function using as an input the total ambient PM_{2.5} concentration. This RR is then used to calculate the total mortalities from PM_{2.5} exposure. At this point, a linear relationship between exposure and RR is not assumed. In addition, the concentration-response function is not used to estimate sector-specific mortalities, instead the fractional contribution of the sector to total PM_{2.5} concentration is used. It seems more that this method assumes a linear dependence between PM_{2.5} emissions and PM_{2.5} concentration. In this study, the percent reductions in PM_{2.5} from each sector sum to 168%. This nonlinearity is what causes the sector-specific mortalities to sum to a larger number than the total PM_{2.5} mortalities.

As an example, if the total mortalities in India were 1 million and you applied the 52% reduction in PM_{2.5} from the residential sector, you get 520,000 deaths. You get the same result regardless of the shape of the concentration-response function that gave you 1 million deaths.

2. The fact that the sector-specific mortality estimates sum to 1.7 million mortalities (in the fractional approach) even though your total mortalities in India is only 1.04 million should be discussed in the text.

3. Similarly, in Table 1, the authors present percent difference reductions in PM concentrations due to switching off one emission sector at a time. The sum of the individual reductions is 168%. The caption in the table notes that this is due to the response of

atmospheric chemistry to removal of emissions. In light of the above discussion, I think this should be made more clear and elaborated on in the main text. What specifically is causing this? I would not have expected the relationship between emissions and PM2.5 to be as superlinear as reported.

4. In lines 176-178, the authors state, "This means that calculating source-specific mortality based on the fractional contribution of each emission sector to total PM2.5 exposure, overestimates sector-specific premature mortality." The fractional contribution approach does overestimate mortality relative the "reduction estimate removing emissions" approach and I agree this is due to the nonlinear relationship of RR and PM2.5. However, I don't necessarily agree that the "reduction estimate" approach is more relevant for estimating sector-specific mortality. The "reduction" approach estimates avoided deaths if emissions from a sector were reduced, and is used in intervention studies, as in Anenberg et al. (2017) (see supplemental) and discussed in Kodros et al. (2017). In Table 2, the "reduction" in mortalities only account for 75% of the total mortalities even though the sectors individually account for 168% of the total PM. Because of the nonlinear concentration-response function this method doesn't attribute all of the PM2.5 mortalities into the different sectors (though some of that remaining fraction may be from sectors not considered here such as sea salt).

Minor comments

The first sentence of the abstract reads "PM2.5 is a leading cause of disease burden in India". I'm not positive on the correct terminology of the health community, but I believe "cause" refers more to disease endpoints while PM2.5 exposure is a "risk factor". Please double check this.

References:

Anenberg, S. C., Henze, D. K., Lacey, F., Irfan, A., Kinney, P., Kleiman, G. and Pillarisetti, A.: Air pollution-related health and climate benefits of clean cookstove programs in Mozambique, *Environ. Res. Lett.*, 12(2), 25006, doi:10.1088/1748-9326/aa5557, 2017.

Kodros, J. K., Wiedinmyer, C., Ford, B., Cucinotta, R., Gan, R., Magzamen, S. and Pierce, J. R.: Global burden of mortalities due to chronic exposure to ambient PM 2.5 from open combustion of domestic waste, *Environ. Res. Lett.*, 11(12), 124022, doi:10.1088/1748-9326/11/12/124022, 2016.

Reviewer #2 (Remarks to the Author):

This paper has changed its focus fairly substantially since the initial submission, placing less emphasis on the estimate of air pollution deaths for all of India, and more emphasis now on the contributions of residential emissions to the concentration and health burden. The authors added new simulations where they evaluate the contributions of several more sectors to the total health burden.

As I stated before, air pollution in India is important for its large influence on health in that region. The authors also estimate air pollution deaths in a way that is supported by the current literature, and can make a good contribution to the literature. However, as before, I am not persuaded that the paper is sufficiently novel to warrant publication in a high-impact journal like Nature Communications. There is a lot that is good about this paper, and it can make a good contribution, but I don't see a strong argument for publishing it in Nature Communications.

The authors have appropriately taken the emphasis off of the estimate of the total mortality burden in India, since that is not highly novel. In estimating the contribution of sectors, then, the authors emphasize that their modeling is done at fine resolution for the first time in India. However, the results of the study – that residential emissions contribute most to the mortality burden – are not different from the previous global studies for India (references 13 and 15). Here the authors should compare their % contributions of sectors with previous studies, but both of the previous studies estimate that residential is most important. Finally the authors emphasize that newly available PM_{2.5} observations in India are used here. But these observations are only used to evaluate the model – while it is good that the authors use the most recently-available observations, these observations are only used for evaluation and they in no way influence the quantitative results, as these are based solely on the model. While Figure 1b is good in that it shows little overall bias, there is a lot of scatter and depending on how the measurement locations align with population, that scatter could translate into errors.

More specific comments:

The authors now appropriately include uncertainty estimates in Table 2 and the abstract. However, these could also be included in Figure 4.

Are there estimates of the contributions of sectors to PM_{2.5} based on chemical mass balance source apportionment, or similar methods? Could those be compared with this model-based study?

l. 1 – consider emphasizing “Health” or “Mortality” in the title rather than “Exposure”.

p. 9 top – The percent contributions of different sectors should be compared with references 13 and 15.

l. 242-245 – The authors describe that they use EDGAR-HTAP, which merges EDGAR emissions with national and regional inventories. For India, does EDGAR-HTAP use EDGAR or some national inventory?

Figure 5 – I would question whether this should be included in the main paper, since it is not the authors' own work, but simply plots the IER functions from reference 30. The caption should say “annual average” and consider showing a range relevant for the study. India is very polluted but I doubt that anyplace has an annual mean of 400 $\mu\text{g m}^{-3}$.

l. 300-303 – Using a single estimate of baseline mortality rates for India is a weakness of the study, given the emphasis on fine-resolution modeling, although state estimates that the authors reference would be impractical. Would it be possible to use state estimates and impose age-specific rates based on national estimates?

- I. 305-6 – Is the “age-specific LE from the standard reference life table” specific for India or a global number?
- I. 314 – I’m not sure what “in quadrature” means, although perhaps that would be apparent to others.
- I. 317 – From what population of values is “2 standard deviations” chosen. There is a single annual average.
- I. 322 – “air pollution experts agree” – I think this language is too strong. No study that I am aware of has yet concluded that different toxicities for different PM2.5 components is supported by the literature.
- I. 324-325 – “if carbonaceous particles are treated the same as sulphates” I don’t understand because that’s what is assumed in this study.

Conibear et al.

Thank you both again for your very helpful comments. Your comments have enabled us to create a much stronger scientific paper and for this we are grateful. We have responded to all comments.

Reviewers' comments are in *italics*.

Our responses are in **bold**. Additions and changes to the manuscript are identified with line numbers.

Reviewer #1 (Remarks to the Author):

Review of Emissions from residential energy use dominate exposure to ambient fine particulate matter in India

The authors estimate premature mortality in India due to exposure to PM_{2.5} from all sources and source-specific mortality rates. The authors include a comparison to surface observations and AOD in India. The revised version of this manuscript focuses on the large contribution of emissions from the residential energy sector to PM_{2.5} and associated premature mortality. The novel aspect of this work is sector-specific mortality estimates over India at a finer model resolution than what is used in typical global models.

Overall, the authors responded to all of my comments and the framing of the revised manuscript is more in line and supported by the results of the study.

I have a few additional concerns that that authors should address before I can recommend publication.

Main comments

1. In lines 50-52, the authors state that scaling the mortalities by the contribution of an emission sector to total ambient PM_{2.5} concentration “assumes a linear relationship between PM_{2.5} exposure and premature mortality”; however, I don’t really agree with this. The relative risk (RR) is calculated

with the nonlinear concentration-response function using as an input the total ambient PM_{2.5} concentration. This RR is then used to calculate the total mortalities from PM_{2.5} exposure. At this point, a linear relationship between exposure and RR is not assumed. In addition, the concentration-response function is not used to estimate sector-specific mortalities, instead the fractional contribution of the sector to total PM_{2.5} concentration is used. It seems more that this method assumes a linear dependence between PM_{2.5} emissions and PM_{2.5} concentration. In this study, the percent reductions in PM_{2.5} from each sector sum to 168%. This nonlinearity is what causes the sector-specific mortalities to sum to a larger number than the total PM_{2.5} mortalities.

As an example, if the total mortalities in India were 1 million and you applied the 52% reduction in PM_{2.5} from the residential sector, you get 520,000 deaths. You get the same result regardless of the shape of the concentration-response function that gave you 1 million deaths.

Thank you for this very important point.

We identified the cause of the discrepancy between population-weighted PM_{2.5} concentrations and mortality estimates from the summation of sectoral contributions (168% of control) as being due to differences in the way we executed different simulations. The control and residential simulations were completed for the original paper submission. The other sectors were added in response to the first round of reviewer comments to strengthen the novelty of our work. However, the original (control and residential) simulations were executed in a different way to the new simulations (minor differences in the way these were submitted to our HPC), which meant that the original simulations were not directly comparable to the new simulations. To correct for this discrepancy between the simulations, we have re-run the original (both the control and residential) simulations in the same way as the other sectors, so all simulations are identical apart from their emissions. This has corrected the original discrepancy and brought the summation of sectoral contributions to PM_{2.5} concentrations to 102% of the control. This change did not alter the estimated contribution of residential energy use emissions to PM_{2.5} concentrations of 52%, though did alter our total premature mortality from 1,038,000 to

990,000 per year. Our overall results and conclusions were not affected by this issue and are unchanged. We thank the reviewer for identifying this issue and we apologise for this mistake. We have modified our paper throughout to account for this change.

We thank the reviewer for pointing out that our description of the different calculations of health impacts was confusing. We have now clarified the different ways we have calculated sector-specific mortality. We have modified the manuscript to describe these two methods (lines 45-55) with a detailed description in the Methods (lines 344-351). We now use the same nomenclature as Kodros et al., (2016) for the two difference methods: subtraction and attribution. The subtraction method calculates the premature mortality from all sources (M_{ALL}) and the premature mortality when one sector has been removed (M_{SECTOR_OFF}). The sector-specific mortality (M_{SECTOR}) is calculated as the difference between M_{ALL} and M_{SECTOR_OFF} :

$$M_{SECTOR} = M_{ALL} - M_{SECTOR_OFF}$$

The attribution method first calculates the sectoral contribution to $PM_{2.5}$ concentrations from removing an emission sector and then uses this fraction to scale the total premature mortality estimate.

$$M_{SECTOR} = M_{ALL} \times (PM_{2.5_ALL} - PM_{2.5_SECTOR_OFF}) / PM_{2.5_ALL}$$

Both methods account for the non-linearities in atmospheric processes when removing emissions on $PM_{2.5}$ concentrations and for the non-linearities in the health function when estimating total premature mortality. However, the subtraction method accounts for the non-linearities in the health function when reducing $PM_{2.5}$ concentrations as oppose to scaling the health impact as in the attribution method. Incremental pollution abatement efforts to reduce health impacts saturate at high $PM_{2.5}$ concentrations (Pope C. A. III et al., (2015)) due to the non-linear exposure-response relationships, which is critically important for heavily polluted regions such as India. Accordingly, the attribution method estimates the number of premature mortalities owing to sector emissions, while the subtraction method estimates the reduction in premature mortalities from removing sector emissions.

Table 2 in the revised paper highlights the differences between these two methods. For the attribution method, the sum of the premature mortality attributed to all the emission sectors is

1,012,000 premature mortalities per year, 102% of the total premature mortality from the control simulation. For the subtraction method, the sum of the premature mortalities for all the emission sectors is 47% of the control. The attribution methods attributes 511,000 premature mortalities per year to residential energy use emissions. The subtraction method suggests that removing residential emissions would prevent 256,000 premature mortalities per year. The premature mortality estimates from the subtraction method are smaller than the attribution method due to the non-linear exposure-response relationship for incremental pollution abatement, where the mortality response to PM_{2.5} concentrations goes sub-linear at the high PM_{2.5} concentrations found over India.

We think this new description is much clearer and we thank the reviewer for their comments.

2. The fact that the sector-specific mortality estimates sum to 1.7 million mortalities (in the fractional approach) even though your total mortalities in India is only 1.04 million should be discussed in the text.

Thank you pointing this out. As mentioned above, now that we have rerun our control simulation the summation of sector contributions from the attribution method is now 102% of the control and not as superlinear as before. As above, we note our estimate of total premature mortality changed by 5% from 1,038,000 to 990,000 per year.

3. Similarly, in Table 1, the authors present percent difference reductions in PM concentrations due to switching off one emission sector at a time. The sum of the individual reductions is 168%. The caption in the table notes that this is due to the response of atmospheric chemistry to removal of emissions. In light of the above discussion, I think this should be made more clear and elaborated on in the main text. What specifically is causing this? I would not have expected the relationship between emissions and PM_{2.5} to be as superlinear as reported.

Thank you for these comments. We have now corrected this issue as described in our responses above. The sum of the individual sectors is now 102% of the control. This small difference is likely due to the response of atmospheric chemistry to removal of emissions.

4. In lines 176-178, the authors state, “This means that calculating source-specific mortality based on the fractional contribution of each emission sector to total PM_{2.5} exposure, overestimates sector-specific premature mortality.” The fractional contribution approach does overestimate mortality relative the “reduction estimate removing emissions” approach and I agree this is due to the nonlinear relationship of RR and PM_{2.5}. However, I don’t necessarily agree that the “reduction estimate” approach is more relevant for estimating sector-specific mortality. The “reduction” approach estimates avoided deaths if emissions from a sector were reduced, and is used in intervention studies, as in Anenberg et al. (2017) (see supplemental) and discussed in Kodros et al. (2017). In Table 2, the “reduction” in mortalities only account for 75% of the total mortalities even though the sectors individually account for 168% of the total PM. Because of the nonlinear concentration-response function this method doesn’t attribute all of the PM_{2.5} mortalities into the different sectors (though some of that remaining fraction may be from sectors not considered here such as sea salt).

Thank you for clarifying this important point. We agree that one method is not better than the other, but help provide different information. We have made our discussion clearer to state the differences between the methods, the different results they produce, the reasons why they produce different results and that they address different questions.

Minor comments

The first sentence of the abstract reads “PM_{2.5} is a leading cause of disease burden in India”. I’m not positive on the correct terminology of the health community, but I believe “cause” refers more to disease endpoints while PM_{2.5} exposure is a “risk factor”. Please double check this.

Thank you for pointing this out. You are correct and we have changed accordingly.

References:

Anenberg, S. C., Henze, D. K., Lacey, F., Irfan, A., Kinney, P., Kleiman, G. and Pillarisetti, A.: Air pollution-related health and climate benefits of clean cookstove programs in Mozambique, *Environ. Res. Lett.*, 12(2), 25006, doi:10.1088/1748-9326/aa5557, 2017.

Kodros, J. K., Wiedinmyer, C., Ford, B., Cucinotta, R., Gan, R., Magzamen, S. and Pierce, J. R.: Global burden of mortalities due to chronic exposure to ambient PM 2.5 from open combustion of domestic waste, *Environ. Res. Lett.*, 11(12), 124022, doi:10.1088/1748-9326/11/12/124022, 2016.

References:

Jin, Q., Yang, Z.-L. & Wei, J. Seasonal Responses of Indian Summer Monsoon to Dust Aerosols in the Middle East, India, and China. *J. Clim.* 29, 6329–6349 (2016).

Pope III, C. A., Cropper, M., Coggins, J. & Cohen, A. Health Benefits of Air Pollution Abatement Policy: Role of the Shape of the Concentration-Response Function. *J. Air Waste Manage. Assoc.* 65, 516–522 (2015).

Reviewer #2 (Remarks to the Author):

This paper has changed its focus fairly substantially since the initial submission, placing less emphasis on the estimate of air pollution deaths for all of India, and more emphasis now on the contributions of residential emissions to the concentration and health burden. The authors added new simulations where they evaluate the contributions of several more sectors to the total health burden.

As I stated before, air pollution in India is important for its large influence on health in that region. The authors also estimate air pollution deaths in a way that is supported by the current literature, and can make a good contribution to the literature. However, as before, I am not persuaded that the paper is sufficiently novel to warrant publication in a high-impact journal like Nature Communications. There is a lot that is good about this paper, and it can make a good contribution, but I don't see a strong argument for publishing it in Nature Communications.

The authors have appropriately taken the emphasis off of the estimate of the total mortality burden in India, since that is not highly novel. In estimating the contribution of sectors, then, the authors emphasize that their modeling is done at fine resolution for the first time in India. However, the results of the study – that residential emissions contribute most to the mortality burden – are not different from the previous global studies for India (references 13 and 15). Here the authors should compare their % contributions of sectors with previous studies, but both of the previous studies estimate that residential is most important. Finally the authors emphasize that newly available PM_{2.5} observations in India are used here. But these observations are only used to evaluate the model – while it is good that the authors use the most recently-available observations, these observations are only used for evaluation and they in no way influence the quantitative results, as these are based solely on the model. While Figure 1b is good in that it shows little overall bias, there is a lot of scatter and depending on how the measurement locations align with population, that scatter could translate into errors.

We thank the reviewer for these comments. With regards to the novelty of our work, we are the first to quantify the sectoral contributions to mortality estimates from exposure to ambient air quality over India using a high-resolution, online and coupled regional model. Recent studies addressing this important question used coarser resolution, offline, global chemical transport models (Lelieveld et al., (2015) and Silva et al., (2016)). The online-coupling between gas, aerosol and meteorology on the same timestep captures feedbacks throughout the atmospheric system and this is important when estimating sectoral contributions through removing emissions. Previous studies were unable to evaluate their PM_{2.5} concentrations due to a lack of observational data. A careful evaluation against recent observations allows us to provide a stronger constraint on our model estimates and lends additional confidence to our results. In summary, our paper is the first high-resolution study of the sector-specific contribution to air pollution mortality in India and the first study to be evaluated against extensive PM_{2.5} observations. We have reworded the Abstract and Conclusions to make this clearer.

We have expanded the comparison of our work against previous studies as suggested by the referee. We have also expanded our analysis of the different factors that alter premature mortality estimates in our study. This analysis highlights the range of factors and assumptions that impact final premature mortality estimates and indicates the likely causes of difference between previous studies. These different factors can act in opposite directions. For example, two studies with different PM_{2.5} concentrations and different exposure-response relationships can estimate similar premature mortality. This discussion is on lines 128-200.

As suggested by the referee, we have also expanded our comparison against previous estimates of sector-specific contributions to premature mortality. This comparison is detailed in Figure 4. This shows that our estimate of the premature mortality attributed to residential energy use emissions is at the upper end of previous studies. This is due to higher PM_{2.5} concentrations in our study (in line with observations) combined with a less sensitive exposure-response relationship.

Our study is the first comparison against extensive PM_{2.5} observations. Overall, the model is relatively unbiased. We agree that there is scatter in the comparison between our model and observations. Our next step will be to attempt to identify model issues and reduce this scatter in future simulations. This is an extended piece of work requiring detailed evaluation of potential causes, and is beyond the scope of the current work.

More specific comments:

The authors now appropriately include uncertainty estimates in Table 2 and the abstract. However, these could also be included in Figure 4.

Thank you for this very good idea. We have added uncertainty ranges to the plot. No uncertainty ranges are plotted for some studies due to lack of data given for all or part of Chafe et al., (2014), Lelieveld et al., (2015), Apte et al., (2015), Chowdhury and Dey (2016) and WHO (2014).

Are there estimates of the contributions of sectors to PM_{2.5} based on chemical mass balance source apportionment, or similar methods? Could those be compared with this model-based study?

This is a useful suggestion. However, a direct estimate is difficult because source apportionment studies differentiate fossil and non-fossil sources of BC. In India, residential energy use emissions will be predominantly from biofuel, but the emissions classification will also include residential fossil sources (e.g. coal), which complicates the comparison. A true comparison would need emission data separately for fossil and non-fossil residential sources. We add the following sentence: “Source apportionment suggest that 46-73% of BC concentrations in India are from non-fossil source (residential biofuel and biomass burning) (Butt et al., (2016), which broadly matches our estimate of the contribution of residential emissions to PM_{2.5} concentrations.”

l. 1 – consider emphasizing “Health” or “Mortality” in the title rather than “Exposure”.

Good idea, we have changed accordingly.

p. 9 top – The percent contributions of different sectors should be compared with references 13 and 15.

Thank you for pointing out the lack of this in our work. The comparison of mortality contributions from emission sectors to previous studies has been added to Figure 4 and the discussion (see lines 180-200).

l. 242-245 – The authors describe that they use EDGAR-HTAP, which merges EDGAR emissions with national and regional inventories. For India, does EDGAR-HTAP use EDGAR or some national inventory?

EDGAR-HTAPv2.2 uses the Model Intercomparison Study for Asia Phase III (MIX), which is a mosaic Asian anthropogenic emission inventory (Li et al., 2017). For India, MIX used the Indian emission inventory provided by Argonne National Laboratory (Lu et al., 2011, Lu and Streets, 2012) for SO₂, BC, and OC for all sectors as well as NO_x for power plants, and REAS2.1 (Kurokawa et al., 2013) for other species.

Figure 5 – I would question whether this should be included in the main paper, since it is not the authors' own work, but simply plots the IER functions from reference 30. The caption should say "annual average" and consider showing a range relevant for the study. India is very polluted but I doubt that anyplace has an annual mean of 400 ug m⁻³.

Thank you for this point. We agree that since the IER functions were derived as part of the GBD2015, we have moved this plot to the supplementary. We have also changed the PM_{2.5} concentration range to suitable annual-means for India. To note these IER functions are different to reference 30, Burnett et al., (2014)., which was used for GBD2010.

l. 300-303 – Using a single estimate of baseline mortality rates for India is a weakness of the study, given the emphasis on fine-resolution modeling, although state estimates that the authors reference would be impractical. Would it be possible to use state estimates and impose age-specific rates based on national estimates?

We have added a sensitivity study using the state-specific baseline mortality rates by Chowdhury and Dey (2016) applied to the newer and age-specific rates from the GBD2015. They applied per capita gross domestic product (GDP) proxies to WHO health statistics from 2011, creating national and state-specific values baseline mortality. To account for variation with age and to utilise the latest data, we calculated the state-variations from the national values and used them to scale the GBD2015 baseline mortalities for COPD, IHD and CEV. This was done for mean, upper and lower confidence intervals. Baseline mortality for LC did not exhibit

any relation with GDP and they did not study ALRI, accordingly we directly use the GBD2015 value for these diseases. Using these state-specific baseline mortalities, our estimate of premature mortality from all sources reduces by 3%.

l. 305-6 – Is the “age-specific LE from the standard reference life table” specific for India or a global number?

The main results use age-specific LE from the GBD2015, which are global values. We have added a sensitivity study using life expectancy data specific to India for 2014 from the Government of India, Ministry of Statistics & Programme Implementation (2016). Using these Indian-specific life expectancy values our value for years of life lost from all sources is 60% lower.

l. 314 – I’m not sure what “in quadrature” means, although perhaps that would be apparent to others.

Fractional uncertainties were added in quadrature (i.e. square root of the sum of squares). We clarify this in the Methods.

l. 317 – From what population of values is “2 standard deviations” chosen. There is a single annual average.

The 2 standard deviations are taken across the domain for the annual mean PM_{2.5} concentrations. We assumed the uncertainties in the PM_{2.5} concentration is represented by the model simulated annual 2 standard deviations for each grid cell as outlined in Lelieveld et al., (2013).

l. 322 – “air pollution experts agree” – I think this language is too strong. No study that I am aware of has yet concluded that different toxicities for different PM2.5 components is supported by the literature.

Thanks for pointing this out, you are correct and we have reworded accordingly.

l. 324-325 – “if carbonaceous particles are treated the same as sulphates” I don’t understand because that’s what is assumed in this study.

Apologies, we were unclear that this statement is a follow on from the previous about how mortality results would change if toxicity were treated differently. We have removed this sentence to avoid confusion.

References:

Apte, J. S., Marshall, J. D., Cohen, A. J. & Brauer, M. Addressing Global Mortality from Ambient PM2.5. *Environ. Sci. Technol.* 49, 8057–8066 (2015).

Burnett, R. T. *et al.* An integrated risk function for estimating the global burden of disease attributable to ambient fine particulate matter exposure. *Environ. Health Perspect.* 122, 397–403 (2014).

Chafe, Z. A. *et al.* Household Cooking with Solid Fuels Contributes to Ambient PM2.5 Air Pollution and the Burden of Disease. *Environ. Health Perspect.* 122, 1314–1320 (2014).

Chowdhury, S. & Dey, S. Cause-specific premature death from ambient PM2.5 exposure in India: Estimate adjusted for baseline mortality. *Environ. Int.* 91, 283–290 (2016).

Cohen, A. J. et al. Estimates and 25-year trends of the global burden of disease attributable to ambient air pollution: An analysis of data from the Global burden of Diseases Study 2015. *Lancet* 389, 1907–1918 (2017).

Kurokawa, J. et al. Emissions of air pollutants and greenhouse gases over Asian regions during 2000–2008: Regional Emission inventory in ASia (REAS) version 2. *Atmos. Chem. Phys.* 13, 11019–11058 (2013).

Lelieveld, J., Barlas, C., Giannadaki, D. & Pozzer, A. Model calculated global, regional and megacity premature mortality due to air pollution. *Atmos. Chem. Phys.* 13, 7023–7037 (2013).

Lelieveld, J. Clean air in the Anthropocene. *Faraday Discuss.* 200, 693–703 (2017).

Li, M. et al. MIX: a mosaic Asian anthropogenic emission inventory under the international collaboration framework of the MICS-Asia and HTAP. *Atmos. Chem. Phys.* 17, 935–963 (2017).

Lu, Z., Zhang, Q. & Streets, D. G. Sulfur dioxide and primary carbonaceous aerosol emissions in China and India, 1996–2010. *Atmos. Chem. Phys.* 11, 9839–9864 (2011).

Lu, Z. & Streets, D. G. Increase in NO_x Emissions from Indian Thermal Power Plants during 1996–2010: Unit-Based Inventories and Multisatellite Observations. *Environ. Sci. Technol.* 46, 7463–7470 (2012).

Ministry of Statistics & Programme Implementation. Life expectancy at birth - 2014. (2016). at <data.gov.in>

WHO. Burden of disease from Ambient Air Pollution for 2012 - Results. 2014 (2014).

Additional improvements

- Updated model evaluation since new control simulation

REVIEWERS' COMMENTS:

Reviewer #1 (Remarks to the Author):

Review of Emissions from residential energy use dominate exposure to ambient fine particulate matter in India

The authors have addressed all of my previous comments and I feel this has improved the manuscript. Overall, I am satisfied with the revisions.

I have one last suggested citation (though I do not require to see the manuscript again). With regard to the "attribution" method, a discussion of this method is included in the GBD MAPS report: <https://www.healtheffects.org/publication/burden-disease-attributable-coal-burning-and-other-air-pollution-sources-china>

Reviewer #2 (Remarks to the Author):

This paper has improved in response to reviewer comments. With exception of some points noted below, I think that the paper is approaching being publishable in general. However, I will repeat my earlier comments that I do not see a compelling reason that this paper should appear in a high-impact journal like Nature Communications. The paper is mainly a step forward beyond other papers – mainly from the resolution of the regional model vs. global models – and the results do not differ very much from those previous papers. I also think that the writing is not as thoughtfully constructed as one would hope for a journal like Nature Communications – some comments are below but it is impossible to call attention to all cases where the writing could be improved.

I first want to discuss a few methodological issues.

1) On p. 12, l. 239, the authors state that the "model meteorology was reinitialized every month". As I understood, they are doing nudging to global meteorology. That nudging should be continuous and it states later that the nudging was done every 3 hours. Why is it then necessary to reinitialize every month? Perhaps I don't understand the procedure sufficiently well.

2) The authors added a sensitivity where they use state-specific baseline mortality rates rather than a national average obtained from GBD2015 (p. 15, l. 323-331). I found it interesting (and perhaps strange) that the authors chose to make the national average results their "base case" and use the state-specific results the sensitivity case. I would expect that the state-specific approach would provide the better estimate. In any case, it seems that the two cases are just 3% different (p.8, l. 146).

3) In estimating YLL, they find that using country-specific life expectancy values give estimates that are 60% lower than using the GBD2015 LE (l. 337). This is a striking

difference that goes without explanation. Again, I wonder which of the two values is more likely to be accurate. I'd like to see more discussion, but fortunately, the YLL values are not a major focus of the paper.

Other comments:

l. 14-15 – I find “constrained by” to be too strong wording. The model results are in no way changed based on the observations.

Abstract – it might be helpful to state the conclusions about the main sources of uncertainty based on the sensitivity / uncertainty analysis done, such as appears in l. 152-155.

l.45 – The paper describes differences in “subtraction” and “attribution” methods. The discussion is good, but I’m not sure these terms are used elsewhere in the literature. An alternative to subtraction would be “zero-out”.

Table 2 – I don’t think the acronyms (AGR etc.) have been defined in the paper before this Table. Also, it is necessary to repeat $\times 10^3$ and (95UI) above each column?

Figure 3 – If every state has RES as being most important, then is it necessary to show panel b? I don’t see what is gained. As an alternative, you could consider plotting either the fraction of deaths in each state from RES (with all numbers >50%), or the deaths per million population.

l. 156-195 – This section focuses on results, showing a lot of results for the “subtraction” vs. “attribution” methods. But I don’t see where the difference between these methods is explained. A short explanation is given in the abstract (l. 20-22) and so I was expecting a somewhat longer discussion of this point in the text.

Figure 4 – consider making 2 figures – one for national mortality estimates and 1 for sectors. Or do a separate right scale for sectors. Also, consider adding a bar for transportation showing the results from Chambliss:

Chambliss, S. E., R. Silva, J. J. West, M. Zeinali, and R. Minjares (2014) Estimating source-attributable health impacts of ambient particulate matter exposure: global premature mortality from surface transportation emissions in 2005, *Environmental Research Letters*, 9, 104009, doi: 10.1088/1748-9326/9/10/104009.

l. 207 – “Evaluation against new observations suggests a large fraction ...” I don’t see how the evaluation shows this. You base these results on your model, and those conclusions may be supported by new observations, but I don’t think that the model evaluation reaches this conclusion.

l.239 – “was reinitialized every month” – you don’t actually say that this was a full annual simulation, nor for which year, until later.

l. 252 – “These studies all...” not clear what studies this refers to.

l. 300-303 – You don’t state the year that the population estimates are for. It would also be helpful to know what national total exposed population was used.

l. 340-342 – It seems there should be some references here talking about the importance of household air pollution for health.

l. 352 – Now “in quadrature” is explained, but I wonder how the results might be different from a Monte Carlo analysis. Doing the in quadrature estimate may be justifiable – I don’t know.

l. 355 – I still don’t think “2 standard deviations” explains what population of values the 2 standard deviations is chosen from.

l. 358 – “defined uncertainty values” – defined by whom? Where?

Conibear et al.

Thank you both again very much for your time and effort in this process. We have responded to all comments.

Reviewers' comments are in *italics*.

Our responses are in **bold**. Additions and changes to the manuscript are identified with line numbers.

Reviewer #1 (Remarks to the Author):

Review of Emissions from residential energy use dominate exposure to ambient fine particulate matter in India

The authors have addressed all of my previous comments and I feel this has improved the manuscript.

Overall, I am satisfied with the revisions.

Thank you. We are grateful for all your help in improving this work.

I have one last suggested citation (though I do not require to see the manuscript again). With regard to the "attribution" method, a discussion of this method is included in the GBD MAPS report: <https://www.healtheffects.org/publication/burden-disease-attributable-coal-burning-and-other-air-pollution-sources-china>

We appreciate you pointing out this important piece of work. We have added the citation of the work to the paper.

Reviewer #2 (Remarks to the Author):

This paper has improved in response to reviewer comments. With exception of some points noted below, I think that the paper is approaching being publishable in general. However, I will repeat my earlier comments that I do not see a compelling reason that this paper should appear in a high-impact journal like Nature Communications. The paper is mainly a step forward beyond other papers – mainly from the resolution of the regional model vs. global models – and the results do not differ very much from those previous papers. I also think that the writing is not as thoughtfully constructed as one would hope for a journal like Nature Communications – some comments are below but it is impossible to call attention to all cases where the writing could be improved.

Thank you. We are grateful for all your comments. We appreciate that the clarity of our writing can be improved and we have strived to do so with this resubmission.

I first want to discuss a few methodological issues.

1) On p. 12, l. 239, the authors state that the “model meteorology was reinitialized every month”. As I understood, they are doing nudging to global meteorology. That nudging should be continuous and it states later that the nudging was done every 3 hours. Why is it then necessary to reinitialize every month? Perhaps I don’t understand the procedure sufficiently well.

Yes, nudging is applied continuously. However, the input data for nudging (i.e. the global model GFS data) has a 3-hour update interval (one data file every 3 hours), and values are interpolated in-between. Nudging is only applied to selected variables and with a nudging coefficient that still allows for WRF-Chem to create its own dynamic. This is intentional, so that fine-scale meteorology within WRF can evolve undisturbed based on the physics and chemistry implemented in WRF-Chem. However, this also means, as with all other forecast models, that the simulation will inevitably drift away from the analysed state. A trade-off is made between allowing WRF-Chem to develop its own internal dynamics and remaining close to the analysed state by nudging during the simulation to keep large-scale dynamics close to GFS analyses and

reinitialising WRF-Chem every so often to force it back to analyses. This is the reason for the “reinitialisation every month”.

2) The authors added a sensitivity where they use state-specific baseline mortality rates rather than a national average obtained from GBD2015 (p. 15, l. 323-331). I found it interesting (and perhaps strange) that the authors chose to make the national average results their “base case” and use the state-specific results the sensitivity case. I would expect that the state-specific approach would provide the better estimate. In any case, it seems that the two cases are just 3% different (p.8, l. 146).

We applied the national data from the GBD2015 so that we were consistent with that study. The GBD2015 baseline national mortality rates were more up-to-date (2015) compared to the state-specific values using data from 2011 and also accounted for the large variations with population age, while the state-specific values did not. We tested the state-specific rates as a sensitivity study (shown as Supplementary Fig. 6) and as the referee points out we found this altered our results by only 3%. We have added a short explanation of this to the corresponding section of the text.

3) In estimating YLL, they find that using country-specific life expectancy values give estimates that are 60% lower than using the GBD2015 LE (l. 337). This is a striking difference that goes without explanation. Again, I wonder which of the two values is more likely to be accurate. I'd like to see more discussion, but fortunately, the YLL values are not a major focus of the paper.

We agree with the referee that this is interesting, but is not a major focus of this paper. We decided to use the GBD2015 normative standard life table for the main YLL results to be consistent with the extensive work done on this issue by the GBD project, and provide YLL results using the India-specific life expectancy in a sensitivity scenario. The GBD project in 2010 developed this normative standard life table after consultation with philosophers, ethicists, and economists to compute YLL at each age by identifying the lowest observed death rate for any age group in countries of more than 5 million in population (Murray et al., (2012) The Lancet).

Accordingly, the discussion of the discrepancies between these underlying value choices is beyond the scope of this work.

Other comments:

l. 14-15 – I find “constrained by” to be too strong wording. The model results are in no way changed based on the observations.

This is a fair point. We have changed “constrained” to “informed”.

Abstract – it might be helpful to state the conclusions about the main sources of uncertainty based on the sensitivity / uncertainty analysis done, such as appears in l. 152-155.

Thanks for this useful comment. We have added the main result from this section to the conclusions. Unfortunately, space restrictions in the Abstract (150 words) prevent us from including this in the abstract.

l.45 – The paper describes differences in “subtraction” and “attribution” methods. The discussion is good, but I’m not sure these terms are used elsewhere in the literature. An alternative to subtraction would be “zero-out”.

This is an important point and we thank the referee for raising it. The term “attribution” was used by Kodros et al., (2016), Chafe et al., (2014), Lelieveld et al., (2015, 2017), HEI GBD MAPS (2016) and Archer-Nicholls et al., (2016). The term “subtraction” was used by Kodros et al., (2016), while Silva et al., (2016) and Chambliss et al., (2014) used “zero-out”. To our knowledge, Kodros et al., (2016) was the only work to use and clarify the differences and implications between both methods and accordingly we decided to be consistent with this work. We add a statement to point out that “zero-out” is an alternative term to “subtraction”.

Table 2 – I don’t think the acronyms (AGR etc.) have been defined in the paper before this Table.

Also, it is necessary to repeat $\times 10^3$ and (95UI) above each column?

We have now defined the sectors earlier in the introduction. We agree it is unnecessary to repeat that information and we have removed it.

Figure 3 – If every state has RES as being most important, then is it necessary to show panel b? I don't see what is gained. As an alternative, you could consider plotting either the fraction of deaths in each state from RES (with all numbers >50%), or the deaths per million population.

Thank you for this helpful comment. We agree that more information could be shown via a different plot. We decided to follow your suggestion of plotting the fraction of premature mortality from residential energy use emissions from both the attribution and substitution method, highlighting the spatial variation in contributions. This provides useful additional information to the reader.

l. 156-195 – This section focuses on results, showing a lot of results for the “subtraction” vs. “attribution” methods. But I don't see where the difference between these methods is explained. A short explanation is given in the abstract (l. 20-22) and so I was expecting a somewhat longer discussion of this point in the text.

The differences between these methods is explained in the introduction (lines 46-56). Implications discussed in the results (lines 171-198) and comparison to previous uses in published work (lines 199-291).

*Figure 4 – consider making 2 figures – one for national mortality estimates and 1 for sectors. Or do a separate right scale for sectors. Also, consider adding a bar for transportation showing the results from Chambliss: Chambliss, S. E., R. Silva, J. J. West, M. Zeinali, and R. Minjares (2014) Estimating source-attributable health impacts of ambient particulate matter exposure: global premature mortality from surface transportation emissions in 2005, *Environmental Research Letters*, 9, 104009, doi: 10.1088/1748-9326/9/10/104009.*

Thanks for this useful suggestion. We have split the figure into total, attributed and substitution estimates to clarify comparisons between this study, sensitivity estimates and previous studies. Thanks for pointing out that important piece of work. We have added it to the figure.

l. 207 – “Evaluation against new observations suggests a large fraction ...” I don’t see how the evaluation shows this. You base these results on your model, and those conclusions may be supported by new observations, but I don’t think that the model evaluation reaches this conclusion.

Thank you for pointing this out. We have clarified that we mean the observed annual mean PM_{2.5} concentrations exceed 100 µg m⁻³ and are well simulated by the model. Then separately from the model we find that 52% of India’s population is exposed to annual mean PM_{2.5} concentrations above 50 µg m⁻³.

l.239 – “was reinitialized every month” – you don’t actually say that this was a full annual simulation, nor for which year, until later.

Thanks for highlighting this lack of clarity. We have added this information earlier within the paper.

l. 252 – “These studies all...” not clear what studies this refers to.

We appreciate you pointing out this out. We have cited to specify what studies we are referring to.

l. 300-303 – You don’t state the year that the population estimates are for. It would also be helpful to know what national total exposed population was used.

We apologise for this and have added this information.

l. 340-342 – It seems there should be some references here talking about the importance of household air pollution for health.

We have added an important reference for household air pollution across India.

l. 352 – Now “in quadrature” is explained, but I wonder how the results might be different from a Monte Carlo analysis. Doing the in quadrature estimate may be justifiable – I don’t know.

Monte Carlo analysis is used in this study as standard to derive the uncertainty ranges in the IER coefficients (lines 340-342), before applying the uncertainty errors in quadrature (similar approach to that used by work on emission inventories over Asia e.g. Kurokawa et al., (2013) and Lu et al., (2011)). Monte Carlo analysis has been used in previous work estimating premature mortality from ambient PM_{2.5} exposure (Chen et al., (2017), Jain et al., (2016), Liu et al., (2009) and Silva et al., (2016)) to propagate total uncertainty in various parameters. However, the uncertainty ranges in these studies remain similarly large at approximately +/- 50% compared to combining in quadrature. Uncertainty ranges are only noticeably reduced when using sophisticated Bayesian Hierarchical modelling as performed by recent updates to the GBD project (GBD2015 (2016), Cohen et al., (2017) and GBD2016 (2017)).

l. 355 – I still don’t think “2 standard deviations” explains what population of values the 2 standard deviations is chosen from.

We have added information in to clarify that this means for each grid cell across the domain we apply ± 2 standard deviations in weekly PM_{2.5} concentrations to the annual mean PM_{2.5} concentrations.

l. 358 – “defined uncertainty values” – defined by whom? Where?

We now specify we are referring to the upper and lower uncertainty intervals of baseline mortality rates provided by the GBD2015 (2016) (available at <http://ghdx.healthdata.org/gbd-results-tool>) and the state-specific (available in the supplementary information of Chowdhury and Dey (2016)).

References

Archer-Nicholls, S. et al. The Regional Impacts of Cooking and Heating Emissions on Air Quality and Disease Burden in China. *Environ. Sci. Technol.* 50, 9416–9423 (2016).

Chafe, Z. A. et al. Household Cooking with Solid Fuels Contributes to Ambient PM_{2.5} Air Pollution and the Burden of Disease. *Environ. Health Perspect.* 122, 1314–1320 (2014).

Chambliss, S. E., Silva, R., West, J. J., Zeinali, M. & Minjares, R. Estimating source-attributable health impacts of ambient fine particulate matter exposure: global premature mortality from surface transportation emissions in 2005. *Environ. Res. Lett.* 9, 104009 (2014).

Chen, L. et al. Assessment of population exposure to PM_{2.5} for mortality in China and its public health benefit based on BenMAP. *Environ. Pollut.* 221, 311–317 (2017).

Chowdhury, S. & Dey, S. Cause-specific mortality from ambient PM_{2.5} exposure in India: Estimate adjusted for baseline mortality Supplementary Information. *Environ. Int.* 91, (2016).

Cohen, A. J. et al. Estimates and 25-year trends of the global burden of disease attributable to ambient air pollution: An analysis of data from the Global burden of Diseases Study 2015. *Lancet* 389, 1907–1918 (2017).

GBD 2015 Risk Factors Collaborators. Supplement to: GBD 2015 Risk Factors Collaborators. Global, regional, and national comparative risk assessment of 79 behavioural, environmental and occupational, and metabolic risks or clusters of risks, 1990–2015: a systematic analysis for the Global Bur. *Lancet* 388, 1659–724 (2016).

GBD 2016 Risk Factors Collaborators. Supplementary Appendix 1. Global, regional, and national comparative risk assessment of 84 behavioural, environmental and occupational, and

metabolic risks or clusters of risks, 1990–2016: a systematic analysis for the Global Burden of Disease Study 2016. *Lancet* 390, (2017).

HEI. Burden of Disease Attributable to Coal-Burning and Other Air Pollution Sources in China. Spec. Rep. 20. GBD MAPS Work. Gr. (2016).

Jain, V., Dey, S. & Chowdhury, S. Ambient PM_{2.5} exposure and premature mortality burden in the holy city Varanasi, India. *Environ. Pollut.* 226, 182–189 (2017).

Kodros, J. K. et al. Global burden of mortalities due to chronic exposure to ambient PM_{2.5} from open combustion of domestic waste. *Environ. Res. Lett.* 11, 1–9 (2016).

Kurokawa, J. et al. Emissions of air pollutants and greenhouse gases over Asian regions during 2000–2008: Regional Emission inventory in ASia (REAS) version 2. *Atmos. Chem. Phys.* 13, 11019–11058 (2013).

Lelieveld, J., Evans, J. S., Fnais, M., Giannadaki, D. & Pozzer, A. The contribution of outdoor air pollution sources to premature mortality on a global scale. *Nature* 525, 367–371 (2015).

Liu, J., Mauzerall, D. L. & Horowitz, L. W. Evaluating inter-continental transport of fine aerosols:(2) Global health impact. *Atmos. Environ.* 43, 4339–4347 (2009).

Lu, Z., Zhang, Q. & Streets, D. G. Sulfur dioxide and primary carbonaceous aerosol emissions in China and India, 1996–2010. *Atmos. Chem. Phys.* 11, 9839–9864 (2011).

Murray, C. J. L. et al. GBD 2010: Design, definitions, and metrics. *Lancet* 380, 2063–2066 (2012).

Silva, R. A., Adelman, Z., Fry, M. M. & West, J. J. The Impact of Individual Anthropogenic Emissions Sectors on the Global Burden of Human Mortality due to Ambient Air Pollution. Environ. Health Perspect. 124, 1776–1784 (2016).

Further edits

We have edited the Supplementary Information to focus on the most relevant figures. Our model evaluation relies on surface PM and AERONET AOD observations. We originally included an additional evaluation against MODIS AOD in the Supplementary Paper (Figs. 15, 16). These figures did not contribute to our overall analysis or the evaluation in the main paper so we have decided to remove them. We are now working on expanding this comparison against satellite AOD using a range of different satellite products.